# Different neuronal populations mediate inflammatory pain analgesia by exogenous and endogenous opioids

Xin-Yan Zhang[1,2†], Yan-Nong Dou[1†], Lei Yuan[1,2], Qing Li[1], Yan-Jing Zhu[1], Meng Wang[1,2], Yan-Gang Sun[1,3*]

[1]Institute of Neuroscience, State Key Laboratory of Neuroscience, CAS Center for Excellence in Brain Science & Intelligence Technology, Chinese Academy of Sciences, Shanghai, China; [2]University of Chinese Academy of Sciences, Beijing, China; [3]Shanghai Center for Brain Science and Brain-Inspired Intelligence Technology, Shanghai, China

**Abstract** Mu-opioid receptors (MORs) are crucial for analgesia by both exogenous and endogenous opioids. However, the distinct mechanisms underlying these two types of opioid analgesia remain largely unknown. Here, we demonstrate that analgesic effects of exogenous and endogenous opioids on inflammatory pain are mediated by MORs expressed in distinct subpopulations of neurons in mice. We found that the exogenous opioid-induced analgesia of inflammatory pain is mediated by MORs in Vglut2[+] glutamatergic but not GABAergic neurons. In contrast, analgesia by endogenous opioids is mediated by MORs in GABAergic rather than Vglut2[+] glutamatergic neurons. Furthermore, MORs expressed at the spinal level is mainly involved in the analgesic effect of morphine in acute pain, but not in endogenous opioid analgesia during chronic inflammatory pain. Thus, our study revealed distinct mechanisms underlying analgesia by exogenous and endogenous opioids, and laid the foundation for further dissecting the circuit mechanism underlying opioid analgesia.

*For correspondence:
yangang.sun@ion.ac.cn

†These authors contributed equally to this work

Competing interests: The authors declare that no competing interests exist.

## Introduction

Opium has been used for thousands of years, and opioid drugs still remain to be the most powerful analgesics used clinically. However, numerous adverse side-effects, including addiction, tolerance, hyperalgesia and respiratory depression, have limited its clinical use (*Colvin et al., 2019*; *Streicher and Bilsky, 2018*; *Trang et al., 2015*). The analgesic effect of morphine-like opioids is mainly mediated by mu-opioid receptor (MOR) (*Kieffer, 1999*; *Matthes et al., 1996*), which is encoded by the *Oprm1* gene with many splicing isoforms (*Chen et al., 1993*; *Pasternak, 2014*; *Wang et al., 1993*). Besides their extensive distribution in the spinal cord and primary sensory neurons (*Kemp et al., 1996*; *Scherrer et al., 2009*), MORs are widely expressed in many pain-related brain regions, including the periaqueductal gray (PAG), thalamus, rostral ventromedial medulla (RVM), and anterior cingulate cortex (ACC) (*Corder et al., 2018*; *Erbs et al., 2015*; *Fields, 2004*). In addition, MORs are highly expressed in the regions which are involved in reward or emotion such as the ventral tegmental area (VTA), nucleus accumbens (NAc), and amygdala (*Fields and Margolis, 2015*; *Lutz and Kieffer, 2013*).

The MORs expressed in different brain areas or different neuronal types might play distinct roles (*Fields and Margolis, 2015*; *Kim et al., 2018*). Activation of MOR by opioid drugs suppresses both sensory and emotional components of pain (*Bushnell et al., 2013*; *Corder et al., 2018*). Pharmacological and genetic approaches have previously been employed to examine the site of action for morphine analgesia (*Maldonado et al., 2018*). Injection of morphine in PAG, RVM and other brain

areas evokes strong inhibition of nocifensive responses (*Manning et al., 1994*; *Yaksh and Rudy, 1977*), and activation of MOR in the PAG induces analgesia by descending modulation of the spinal cord via RVM (*Basbaum and Fields, 1984*). Moreover, opioids can differentially modulate subsets of the RVM neurons, which gate the spinal circuit for nociception via presynaptic mechanisms (*Fields, 2004*; *François et al., 2017*). MOR is also widely expressed in the dorsal root ganglion (DRG) and different neuronal populations at the spinal level (*Kemp et al., 1996*; *Scherrer et al., 2009*; *Spike et al., 2002*; *Wang et al., 2018*). Previous studies suggest that these MORs play important roles in mediating morphine-induced antinociception in both acute and inflammatory pain (*Corder et al., 2017*; *Stein et al., 2009*; *Sun et al., 2019*; *Weibel et al., 2013*). Thus, the site of action for analgesia by systemic morphine administration remains incompletely resolved.

Endogenous opioid system also plays a critical role in gating neural circuits underlying nociceptive information processing, as evidenced by elevated pain in response to the opioid antagonist (*Sun et al., 2003*). The important role of endogenous opioid system is also supported by the finding that persistent pain and placebo induced significant activation of opioid system (*Wager et al., 2007*; *Zubieta et al., 2005*), and placebo-induced analgesia was reversed by non-selective opioid antagonist naloxone (*Benedetti et al., 1999*). Analgesia by the endogenous opioid system is likely due to the fast release of endogenous opioid peptides in both spinal and supraspinal brain areas, leading to attenuation of sensory and pain-specific affective responses by activating the opioid receptors (*Corder et al., 2018*). In addition, opioid-independent constitutive MOR activation observed in chronic inflammatory pain represents another analgesic mechanism of the opioid system (*Corder et al., 2013*). Although brain imaging studies showed that exogenous and endogenous opioids may activate similar brain areas (*LaGraize et al., 2006*; *Zubieta et al., 2001*), it remains unknown whether their effects are mediated by the same population of neurons.

Elucidating the neural mechanism underlying opioid analgesia is essential for developing novel strategies for pain management. In this study, we determined the identity of MOR$^+$ neurons in multiple brain areas, and delineated the different mechanisms underlying the analgesic effects of exogenous and endogenous opioids, using a combination of genetic and pharmacological approaches.

## Results

### Characterization of MOR$^+$ neurons in the mouse brain

MORs are widely expressed in the peripheral and central nervous system. Previous studies have examined the distribution of MORs in the nervous system with ligand binding-based autoradiography and knock-in reporter mouse lines (*Erbs et al., 2015*; *Gardon et al., 2014*; *Mansour et al., 1988*; *Wang et al., 2018*), or based on the distribution of *Oprm1* mRNA, which encodes the MOR (*George et al., 1994*; *Mansour et al., 1995*). However, the identity of MOR$^+$ neurons in multiple brain regions is still not clear. Thus, we determined the identity of the MOR$^+$ neurons in the brain by examining the expression of glutamatergic and GABAergic neuronal markers in MOR$^+$ neurons. We performed triple fluorescence *in situ* hybridization (FISH) with *Oprm1* (MOR), *Slc17a6* (Vglut2), *Slc32a1* (Vgat) probes in wild-type mice (*Figure 1a–f*, *Figure 1—figure supplement 1a–b*). Consistent with previous studies (*Erbs et al., 2015*; *Mansour et al., 1995*; *Wang et al., 2018*), *Oprm1* was detected in many brain areas. By analyzing the co-localization between *Oprm1* and Vglut2/Vgat, we found that in the central and medial part of thalamus and parabrachial nucleus (PBN), where neurons are mostly glutamatergic, nearly all *Oprm1*$^+$ neurons were Vglut2$^+$ (*Figure 1a,b,d,e and g*). In contrast, most *Oprm1*$^+$ neurons were Vgat$^+$ in the striatum and central amygdala (CeA), which contain mostly GABAergic neurons (*Figure 1a,c and g*). For other brain areas, *Oprm1* was expressed in both glutamatergic and GABAergic neurons, such as the periaqueductal gray (PAG), $22.5 \pm 4.0\%$ of *Oprm1*$^+$ neurons were *Vgat*$^+$ and $73.3 \pm 2.7\%$ of *Oprm1*$^+$ neurons were Vglut2$^+$ (*Figure 1d,f and g*). For the cortex, which expresses low level of Vglut2, we used *Slc17a7* (Vglut1) as a marker for glutamatergic neurons, and found that $78.8 \pm 2.0\%$ of *Oprm1*$^+$ neurons were positive for Vglut1 and only $19.2 \pm 2.2\%$ of *Oprm1*$^+$ neurons were positive for Vgat, suggesting that most cortical *Oprm1*$^+$ neurons were glutamatergic (*Figure 1—figure supplement 1c–f*). In rare cases, such as dorsal raphe nucleus, *Oprm1*$^+$ neurons were neither Vglut2$^+$ nor Vgat$^+$ (*Figure 1d*). Taken together, these results demonstrate that MORs are widely expressed in both glutamatergic and GABAergic neurons in the brain.

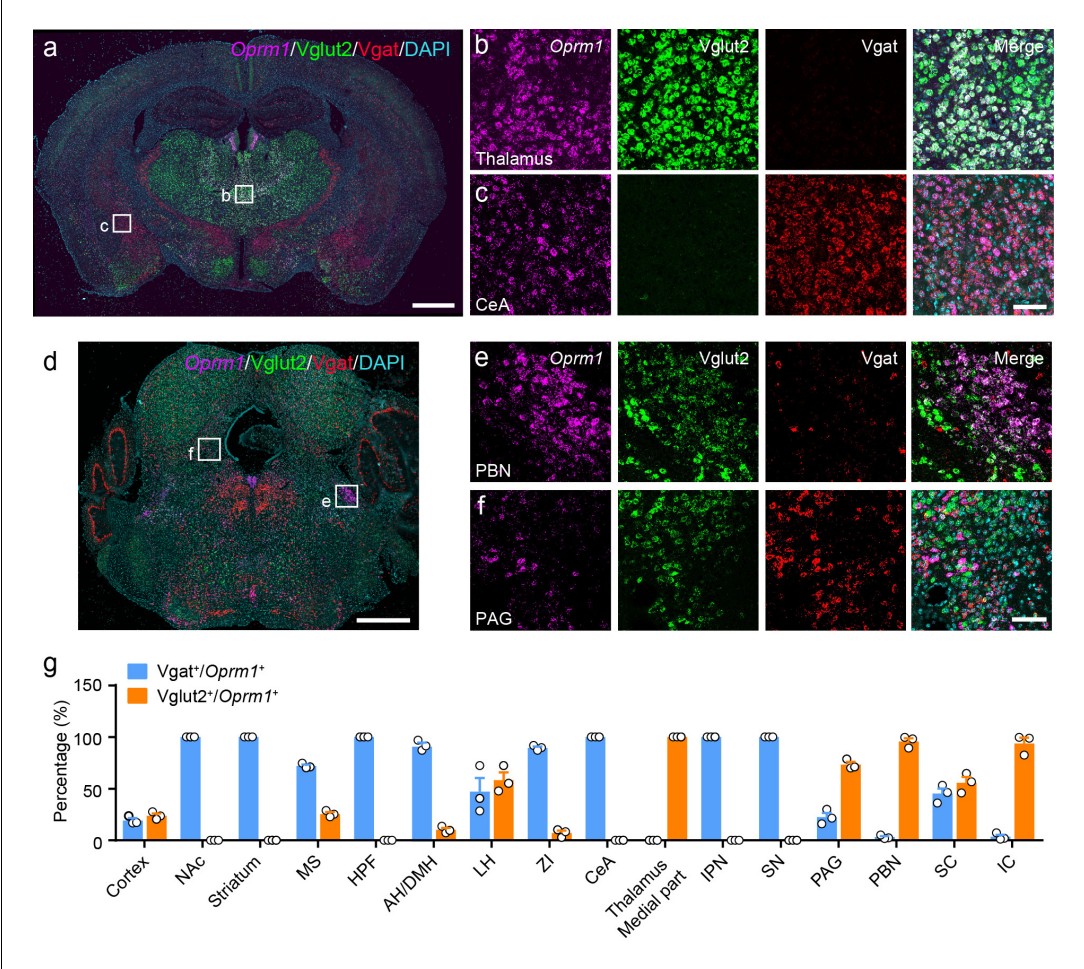

**Figure 1.** Distribution of *Oprm1* in Vglut2+ and Vgat+ neurons in the mouse brain. (**a–f**) Multiple *in situ* hybridization in wild-type mice shows the distribution of *Oprm1* in glutamatergic and GABAergic neurons using RNAscope assay. Scale bars, 1 mm (**a and d**), 100 μm (**b**), (**c**), (**e**), and (**f**). (**g**) Percentage of glutamatergic and GABAergic neurons in MOR expressing neurons in representative brain regions (n = 3 mice). NAc, nucleus accumbens; MS, medial septal nucleus; HPF, hippocampal formation; AH/DMH, anterior hypothalamic area/dorsomedial hypothalamic nucleus; LH, lateral hypothalamic area; ZI, zona incerta; CeA, central amygdala; IPN, interpeduncular nucleus; SN, substantia nigra; PAG, periaqueductal gray; PBN, parabrachial nucleus; SC, superior colliculus; IC, inferior colliculus. Data are presented as mean ± SEM.
The online version of this article includes the following source data and figure supplement(s) for figure 1:

**Source data 1.** Percentage of glutamatergic and GABAergic neurons in MOR expressing neurons in representative brain regions.
**Figure supplement 1.** Multiple *in situ* hybridization in wild-type mice.

## Generation and verification of the *Oprm1*^KI/KI mouse line

To study the functional role of MORs expressed in distinct neuronal populations in opioid analgesia, we generated a mouse line with a strategy similar to the 'knockout first' approach (*Skarnes et al., 2011*). This mouse line allows for selectively expressing MORs in distinct neuronal populations. This was achieved by inserting one stop cassette flanked by two *loxP* sites between exon 1 and exon 2 of *Oprm1* gene (*Figure 2a*). This mouse line, referred to as *Oprm1*^KI/KI hereafter, enables re-expression of MORs under the control of Cre recombinase. Immunostaining results showed that MOR expression was abolished in the brain, spinal cord and dorsal root ganglion (DRG) of *Oprm1*^KI/KI mice compared to wild-type mice (*Figure 2b–c*). Next, we determined whether *Oprm1*^KI/KI mice exhibit deficiency in MOR functions by testing the effects of morphine on pain and locomotion. We found that *Oprm1*^KI/KI mice exhibited no response to morphine in pain and locomotion tests, whereas wide-type littermates exhibited strong morphine-induced analgesia and hyper-locomotion (*Figure 2d–h*). MORs are also known to mediate analgesic effect of endogenous opioids in complete

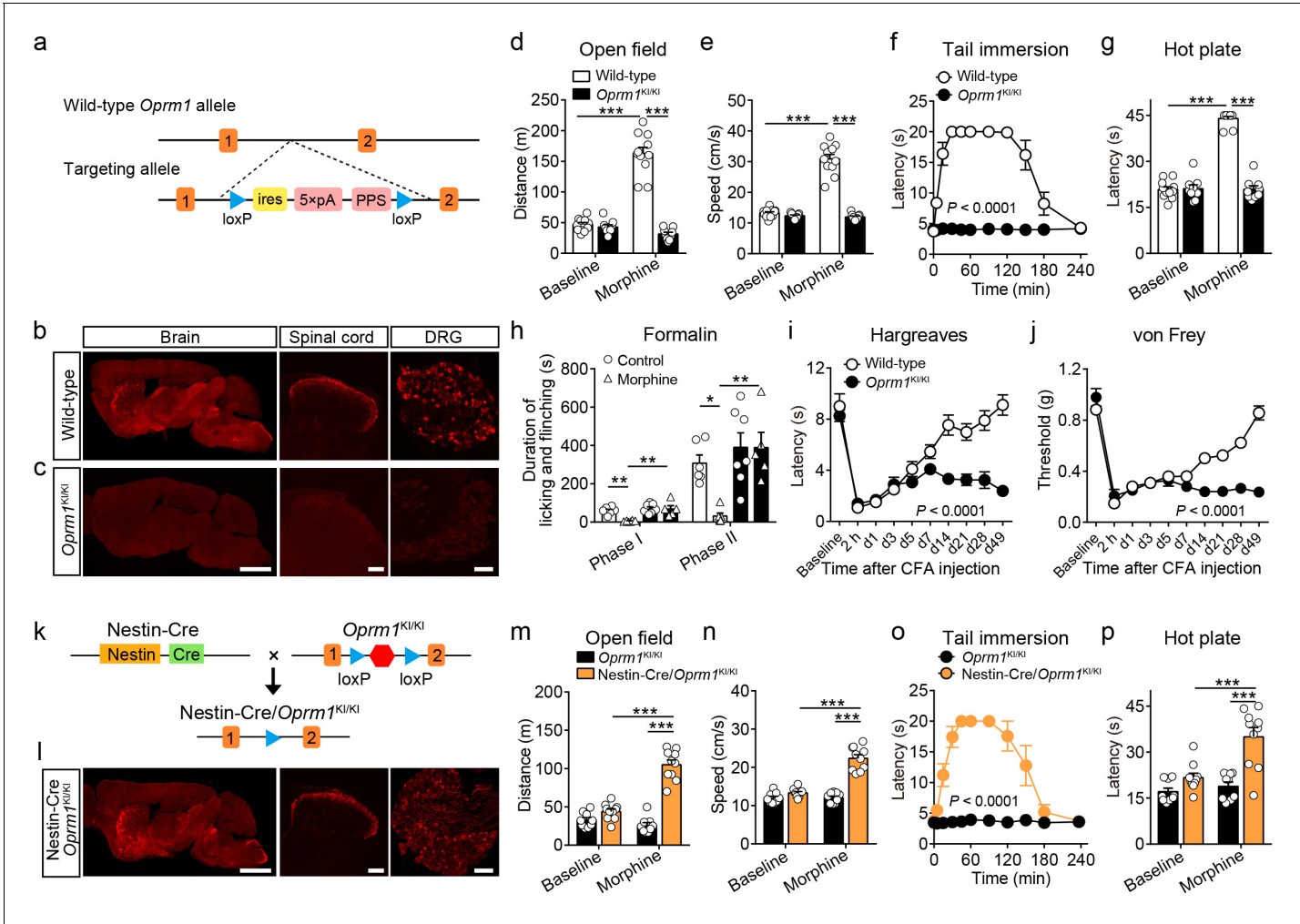

**Figure 2.** Generation and verification of the *Oprm1*[KI/KI] mouse line. (a) Targeting strategy for generating the *Oprm1*[KI/KI] mouse line through inserting a stop cassette flanked by two *loxP* sites between *Oprm1* exon 1 and exon 2. PPS, RNA polymerase pause site. (b and c) MOR expression in brain, spinal cord and DRG sections of wild-type and *Oprm1*[KI/KI] mice. Scale bars, 2 mm (left), 100 μm (middle and right). (d and e) Effects of morphine (10 mg/kg, subcutaneous, s.c.) on locomotor activity in open field test for *Oprm1*[KI/KI] mice (n = 9) compared with wild-type (n = 11) littermates. Two-way ANOVA (d), $F_{1,36}$ = 102.1, p<0.0001; (e), $F_{1,36}$ = 107.3, p<0.0001 with Bonferroni correction. (f and g) Effects of morphine (10 mg/kg, s.c.) on thermal pain tested with tail immersion (50°C) and hot plate (52°C) tests in *Oprm1*[KI/KI] mice compared with wild-type littermates. n = 7–11 mice. Two-way ANOVA (f), $F_{10,132}$ = 36.01, p<0.0001; (g), $F_{1,36}$ = 145.0, p<0.0001 with Bonferroni correction. (h) Summary of the formalin-induced behavioral responses in phase I (0–10 min) and phase II (10–60 min) in *Oprm1*[KI/KI] mice and wild-type littermates treated with saline or morphine (10 mg/kg, intraperitoneal, i.p.). n = 5–7 mice. One-way ANOVA (Phase I, $F_{3,20}$ = 10.10, p=0.0003; Phase II, $F_{3,20}$ = 8.085, p=0.0010) with Bonferroni correction. (i and j) Thermal and mechanical pain tests during complete Freund's adjuvant (CFA)-induced inflammatory pain in *Oprm1*[KI/KI] (n = 6) and wild-type (n = 9) littermates. Two-way ANOVA (i), $F_{9,130}$ = 8.312, p<0.0001; (j), $F_{9,130}$ = 17.90, p<0.0001 with Bonferroni correction. (k) Schematic diagram for driving MOR re-expression. Nestin-Cre mice were crossed with *Oprm1*[KI/KI] mice, and the stop cassette (red hexagon) was excised. (l) MOR expression in the brain, spinal cord and DRG sections of Nestin-Cre/*Oprm1*[KI/KI] mice. Scale bars, 2 mm (left), 100 μm (middle and right). (m and n) Effects of morphine (10 mg/kg, s.c.) on moving distance and average speed of locomotion with open field test in Nestin-Cre/*Oprm1*[KI/KI] mice (n = 10) compared to *Oprm1*[KI/KI] (n = 9) littermates. Two-way ANOVA (m), $F_{1,34}$ = 61.95, p<0.0001; (n), $F_{1,34}$ = 51.10, p<0.0001 with Bonferroni correction. (o and p) Effects of morphine (10 mg/kg, s.c.) on thermal pain tested with tail immersion (50°C) and hot plate (52°C) tests in Nestin-Cre/*Oprm1*[KI/KI] mice, compared to *Oprm1*[KI/KI] littermates. n = 5–10 mice. Two-way ANOVA (o), $F_{10,99}$ = 16.66, p<0.0001; (p), $F_{1,34}$ = 8.616, p=0.0059 with Bonferroni correction. Data are presented as mean ± SEM, *p<0.05, **p<0.01, ***p<0.001.

The online version of this article includes the following source data and figure supplement(s) for figure 2:

**Source data 1.** Raw data of the behavioral tests and MOR expression level in wild-type and Nestin-Cre/Oprm1-KI mice.
**Figure supplement 1.** MOR expression in Nestin-Cre/*Oprm1*[KI/KI] mice.

Freund's adjuvant (CFA)-induced chronic inflammatory pain (*Corder et al., 2013*). Consistent with previous findings, we found that wild-type mice, but not *Oprm1*[KI/KI] mice, recovered from CFA-induced hyperalgesia 3–4 weeks after CFA injection, suggesting that the endogenous opioid analgesia was abolished in *Oprm1*[KI/KI] mice (*Figure 2i–j*). By contrast, *Oprm1*[KI/KI] mice exhibited comparable basal locomotor activity and pain threshold with wild-type littermates (*Figure 2d–j*). These results demonstrate that insertion of a stop cassette in the first intron of *Oprm1* gene abolished expression and function of MORs.

Next, we determined whether MORs could be re-expressed after removal of the stop cassette by Cre recombinase. We bred *Oprm1*[KI/KI] mice with Nestin-Cre mice (*Figure 2k*), which express Cre recombinase in the nervous system under the control of *Nestin* promoter (*Tronche et al., 1999*). Immunostaining showed that the expression pattern of MOR in the brain, spinal cord and DRG of Nestin-Cre/*Oprm1*[KI/KI] mice was comparable to that in wild-type mice (*Figure 2b and l*). We analyzed the expression level of MOR in sections of Nestin-Cre/*Oprm1*[KI/KI] mice, and found the expression of MOR in Nestin-Cre/*Oprm1*[KI/KI] mice was comparable to that in wild-type mice (*Figure 2— figure supplement 1*). Whether re-expressing MOR in Nestin-Cre/*Oprm1*[KI/KI] mice could restore MOR functions was further examined with behavioral experiments. We found that morphine significantly increased the response latency to noxious stimuli and locomotor activity in Nestin-Cre/*Oprm1*[KI/KI] but not in *Oprm1*[KI/KI] mice (*Figure 2m–p*). These data confirmed that the *Oprm1*[KI/KI] mouse line enabled restoration of functional MORs in the presence of Cre recombinase. This mouse line was used throughout this study to examine the role of MORs in different neuronal populations during opioid analgesia.

## MORs in Vglut2[+] glutamatergic neurons mediate exogenous opioid analgesia

Although MORs are highly expressed in the glutamatergic neurons of pain-related regions, the functional role of these MORs in opioid analgesia remains largely unknown. We examined the functional role of MORs expressed in glutamatergic neurons in opioid analgesia with Vglut2-Cre/*Oprm1*[KI/KI] mice (*Figure 3a*), which selectively express MORs in Vglut2[+] glutamatergic neurons. Immunostaining showed that the expression of MOR in several brain areas composed mostly by glutamatergic neurons, such as the habenula, thalamus and PBN, in Vglut2-Cre/*Oprm1*[KI/KI] mice resembled that in wild-type mice (*Figure 3b–c*, *Figure 3—figure supplement 1a–b*). We next examined the functional role of MORs expressed in Vglut2[+] glutamatergic neurons in opioid analgesia with Vglut2-Cre/*Oprm1*[KI/KI] mice. We found that systemic morphine injection induced significant analgesic effect in Vglut2-Cre/*Oprm1*[KI/KI] mice during tail immersion, hot plate, and von Frey tests (*Figure 3d–f*). While, the basal pain thresholds were indistinguishable between Vglut2-Cre/*Oprm1*[KI/KI] and *Oprm1*[KI/KI] littermates (*Figure 3—figure supplement 1c–e*). Further, we determined the role of MORs on Vglut2[+] glutamatergic neurons in mediating morphine analgesia for inflammatory pain. Although the nocifensive behaviors in formalin test were comparable between Vglut2-Cre/*Oprm1*[KI/KI] and *Oprm1*[KI/KI] mice (*Figure 3—figure supplement 1f–g*), Vglut2-Cre/*Oprm1*[KI/KI] mice showed significantly less formalin-induced licking and flinching behaviors as compared with *Oprm1*[KI/KI] mice after morphine treatment (*Figure 3g–h*), suggesting an important role of MORs expressed in Vglut2[+] glutamatergic neurons in mediating morphine analgesia on inflammatory pain. Consistently, under chronic inflammatory pain condition induced by CFA, systemic morphine injection significantly increased the withdrawal latency in Hargreaves test and mechanical threshold in von Frey test in Vglut2-Cre/*Oprm1*[KI/KI] mice but not in *Oprm1*[KI/KI] mice (*Figure 3i–j*), indicating that analgesic effects of morphine on CFA-induced chronic inflammatory pain was restored by selective expressing MOR in Vglut2[+] glutamatergic neurons. Furthermore, locomotion analysis showed that morphine injection significantly increased the locomotor distance and speed in Vglut2-Cre/*Oprm1*[KI/KI] mice but not in *Oprm1*[KI/KI] mice after the morphine treatment (*Figure 3—figure supplement 1h–i*), suggesting the increase of latency in Hargreaves test and mechanical threshold in von Frey test was not due to deficit in motor function. Thus, activation of MORs in Vglut2[+] glutamatergic neurons by systemic morphine suppressed both acute and chronic inflammatory pain.

We next asked whether MORs expressed in Vglut2[+] glutamatergic neurons are also essential for endogenous opioid analgesia. We employed the CFA-induced chronic inflammatory pain model, in which the endogenous opioid analgesia is mediated by MOR (*Figure 2i–j*). We found that both Vglut2-Cre/*Oprm1*[KI/KI] and *Oprm1*[KI/KI] mice developed comparable thermal and mechanical

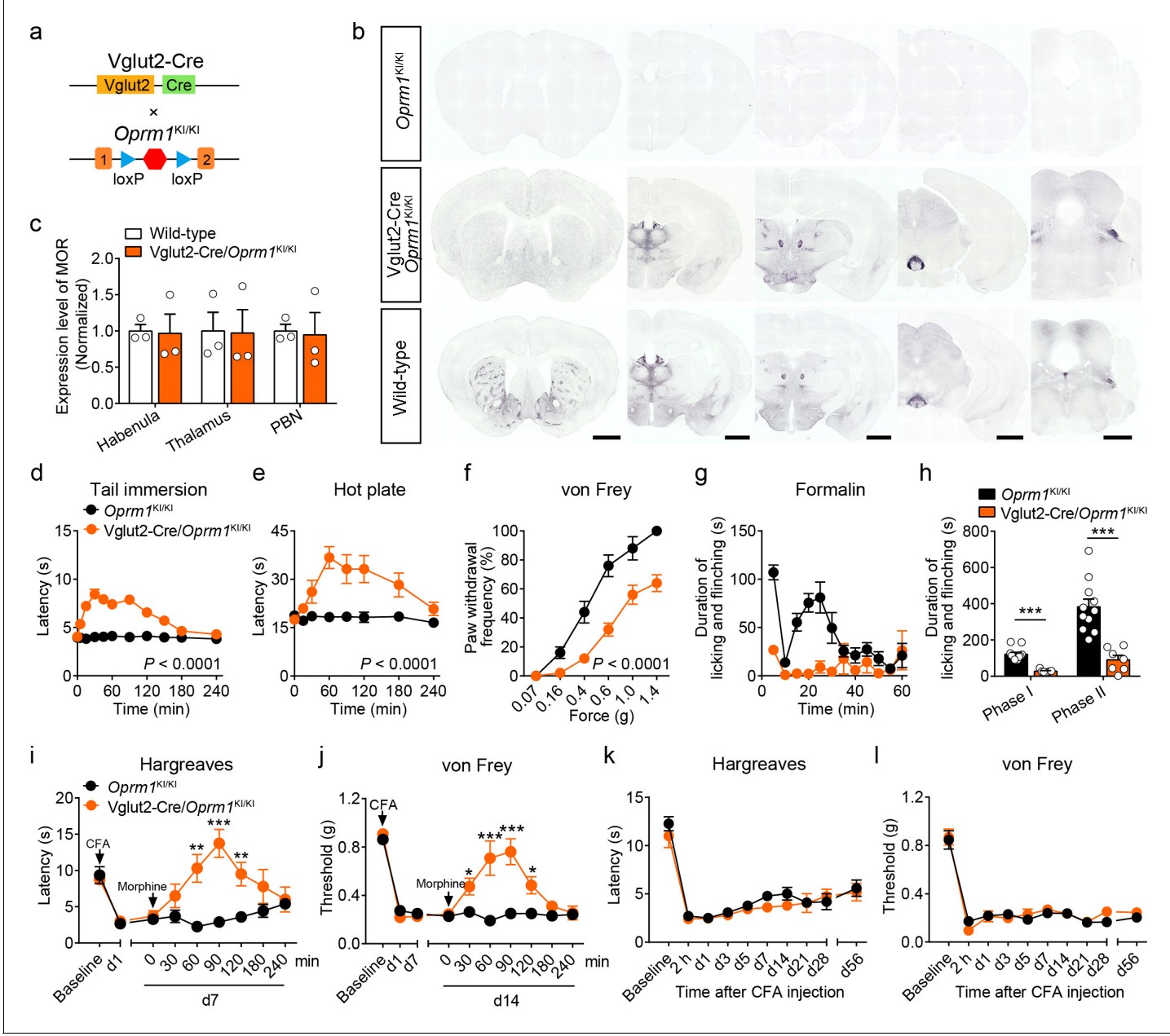

**Figure 3.** MORs in Vglut2[+] glutamatergic neurons mediate exogenous opioid analgesia. (**a**) Schematic diagram for re-expression of MORs in glutamatergic neurons. Vglut2-Cre mice were crossed with *Oprm1*[KI/KI] mice to generate Vglut2-Cre/*Oprm1*[KI/KI] mice. Red hexagon: Stop cassette. (**b**) Graphs showing the immunostaining of MOR in brain sections of *Oprm1*[KI/KI], Vglut2-Cre/*Oprm1*[KI/KI] and wild-type mice. Scale bars, 1 mm. (**c**) Expression level of MOR in Vglut2-Cre/*Oprm1*[KI/KI] mice compared to wild-type mice determined by quantification of signals of MOR by immunostaining. The expression level of MOR was normalized to the mean gray value of MOR signal in each brain region of wild-type mice. (**d–f**) Effects of morphine (10 mg/kg, s.c.) on pain tests with tail immersion (50˚C), hot plate (52˚C) and von Frey tests in *Oprm1*[KI/KI] mice (n = 5–12) and Vglut2-Cre/*Oprm1*[KI/KI] mice (n = 8–10). Two-way ANOVA (**d**), $F_{10,220} = 19.87$, p<0.0001; (**e**), $F_{7,120} = 4.045$, p=0.0005; (**f**), $F_{5,78} = 4.956$, p=0.0005 with Bonferroni correction. (**g**) Time course of formalin-induced nocifensive behaviors in *Oprm1*[KI/KI] (n = 11) and Vglut2-Cre/*Oprm1*[KI/KI] (n = 7) mice with subcutaneous morphine injection (10 mg/kg). (**h**) Summary of the nocifensive behaviors in phase I (0–10 min) and phase II (10–60 min) of formalin test in *Oprm1*[KI/KI] mice (n = 11) and Vglut2-Cre/*Oprm1*[KI/KI] mice (n = 7) with subcutaneous morphine injection (10 mg/kg). Student's unpaired *t*-test, $t_{16} = 6.767$, p<0.0001 (Phase I); $t_{16} = 5.017$, p=0.0001 (Phase II). (**i and j**) Time-course effects of morphine (10 mg/kg, s.c.) on thermal and mechanical sensitivities after CFA application in *Oprm1*[KI/KI] mice and Vglut2-Cre/*Oprm1*[KI/KI] mice on day 7 (**d7**) and d14, respectively. n = 7 mice for each group. Student's unpaired *t* test, $t_{12} = 4.165$, p=0.0013 (**i**), 60 min; $t_{12} = 5.371$, p=0.0002 (**i**), 90 min; $t_{12} = 3.430$, p=0.005 (**i**), 120 min; $t_{12} = 4.273$, p=0.0011 (**j**), 30 min; $t_{12} = 4.110$, p=0.0014 (**j**), 60 min; $t_{12} = 7.365$, p<0.0001 (**j**), 90 min; $t_{12} = 3.983$, p=0.0018 (**j**), 120 min; $t_{12} = 5.517$, p=0.0001 (**j**), 180 min. (**k and l**) Thermal and mechanical pain tests during CFA-induced inflammatory pain in *Oprm1*[KI/KI] and Vglut2-Cre/*Oprm1*[KI/KI] mice. n = 7–9 mice. Two-way

*Figure 3 continued on next page*

*Figure 3 continued*

ANOVA (k), $F_{9,140} = 0.7866$, p=0.9075; (l), $F_{9,140} = 0.4463$, p=0.6291 with Bonferroni correction. Data are presented as mean ± SEM, *p<0.05, **p<0.01, ***p<0.001.

The online version of this article includes the following source data and figure supplement(s) for figure 3:

**Source data 1.** Raw data of the behavioral tests and MOR expression level in Vglut2-Cre/Oprm1-KI and Vglut2-Cre/Oprm1-fl groups of mice.

**Figure supplement 1.** Expression pattern of the re-expressed MOR and functional role of MORs in Vglut2[+] glutamatergic neurons in morphine analgesia.

**Figure supplement 2.** Functional role of MORs in glutamatergic neurons in opioid analgesia.

hyperalgesia after CFA injection (*Figure 3k–l*). Although wild-type mice recovered from CFA-induced chronic inflammatory pain 3–4 weeks after CFA injection (*Figure 2i–j*), neither Vglut2-Cre/*Oprm1*[KI/KI] nor *Oprm1*[KI/KI] mice recovered from CFA-induced hyperalgesia even 8 weeks after CFA injection (*Figure 3k–l*), indicating that analgesia by endogenous opioids is not mediated by MORs in Vglut2[+] glutamatergic neurons. Thus, these results demonstrate that MORs in Vglut2[+] glutamatergic neurons play different roles in exogenous and endogenous opioid analgesia.

We further confirmed these results by selectively deleting MORs from Vglut2[+] glutamatergic neurons (*Figure 3—figure supplement 2a*). Compared with *Oprm1*[fl/fl] mice, Vglut2-Cre/*Oprm1*[fl/fl] mice exhibited selective loss of MOR expression in glutamatergic neuron-enriched regions, including the thalamus and PBN (*Figure 3—figure supplement 2b*). Consistent with the results obtained above, we found that analgesic effects of systemic morphine were significantly decreased in Vglut2-Cre/*Oprm1*[fl/fl] mice in tail immersion and hot plate tests compared with the *Oprm1*[fl/fl] littermates (*Figure 3—figure supplement 2c–d*). For the von Frey test, morphine induced hypersensitivity in Vglut2-Cre/*Oprm1*[fl/fl] mice (*Figure 3—figure supplement 2e*), but analgesia in *Oprm1*[fl/fl] mice (*Figure 3—figure supplement 2f*). Furthermore, in the CFA-induced chronic inflammatory pain model, Vglut2-Cre/*Oprm1*[fl/fl] mice exhibited comparable time course of recovery from hyperalgesia with that of *Oprm1*[fl/fl] littermates (*Figure 3—figure supplement 2g–h*), further confirming that MORs in Vglut2[+] glutamatergic neurons are negligible for analgesia by endogenous opioids. Together, these results indicate that MORs expressed in Vglut2[+] glutamatergic neurons play an important role in mediating morphine analgesia in both acute and inflammatory pain, but not the endogenous opioid analgesia.

## MORs in GABAergic neurons mediate endogenous opioid analgesia

MORs are also highly expressed in the GABAergic neurons. Although recent studies explored the role of MORs expressed in forebrain GABAergic neurons in morphine analgesia (*Charbogne et al., 2017*; *Cui et al., 2014*), the functional role of MORs in GABAergic neurons in opioid analgesia, especially endogenous opioid analgesia, remains largely unknown. We thus examined the role of MORs in GABAergic neurons using Vgat-Cre/*Oprm1*[KI/KI] mice (*Figure 4a*), which selectively express MORs in GABAergic neurons. We found that Vgat-Cre/*Oprm1*[KI/KI] mice showed strong expression of MORs in the striatum, hypothalamus, amygdala, interpeduncular nucleus (IPN) and ventral pallidum (VP) (*Figure 4b* and *Figure 4—figure supplement 1a–b*), consistent with the abundant *Oprm1* expression in GABAergic neurons of these brain areas (*Figure 1g*). Further analysis showed that the expression of MOR in multiple brain areas, such as the striatum, nucleus accumbens (NAc) and CeA, of Vgat-Cre/*Oprm1*[KI/KI] mice resembled that in wild-type mice (*Figure 4c*). Next, we performed behavioral experiments to examine the functional role of MORs expressed in GABAergic neurons with Vgat-Cre/*Oprm1*[KI/KI] mice. The basal nociceptive responses showed no significant difference between Vgat-Cre/*Oprm1*[KI/KI] and *Oprm1*[KI/KI] mice (*Figure 4—figure supplement 1c–e*). However, we found that morphine treatment significantly elevated the response latency of Vgat-Cre/*Oprm1*[KI/KI] mice in the tail immersion test (*Figure 4d*), while systemic administration of morphine resulted in no significant analgesic effect on Vgat-Cre/*Oprm1*[KI/KI] or *Oprm1*[KI/KI] mice in the hot plate test (*Figure 4e*). We next examined the effect of morphine on mechanical pain using von Frey test, and found that systemic morphine injection increased the paw withdrawal frequency in response to mechanical stimulation, indicating that activation of MORs in GABAergic neurons by morphine induced allodynia (*Figure 4f*). To further confirm the results obtained with Vgat-Cre/*Oprm1*[KI/KI] mice, we employed the Vgat-Cre/*Oprm1*[fl/fl] mice (*Figure 4—figure supplement 2a*),

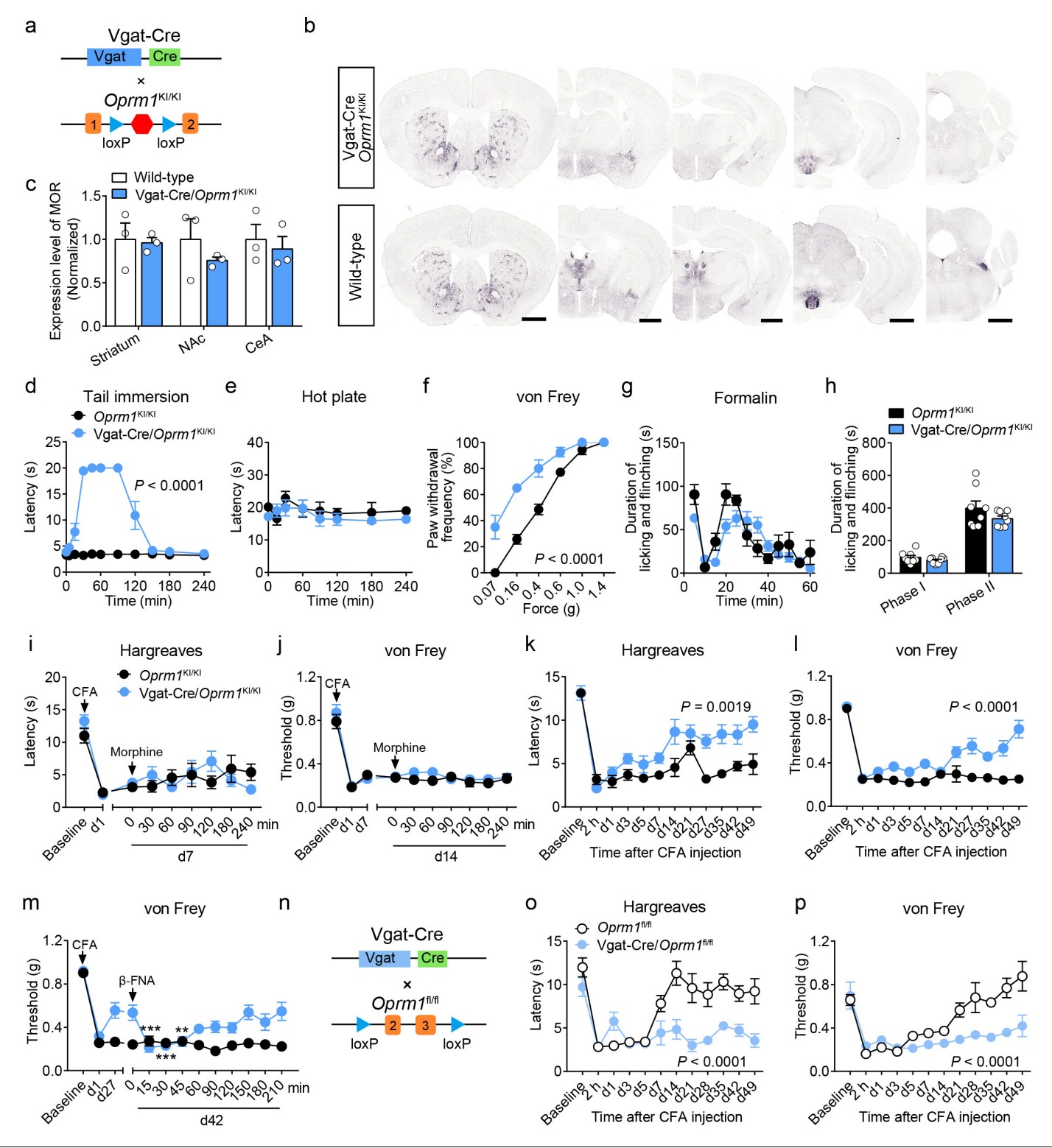

**Figure 4.** MORs in GABAergic neurons mediate endogenous opioid analgesia. (**a**) Schematic diagram for re-expression of MORs in GABAergic neurons. Vgat-Cre mice were crossed with *Oprm1*$^{KI/KI}$ mice to generate Vgat-Cre/*Oprm1*$^{KI/KI}$ mice. Red hexagon: Stop cassette. (**b**) Graphs showing the immunostaining of MOR in Vgat-Cre/*Oprm1*$^{KI/KI}$ and wild-type mouse brain sections. Scale bars, 1 mm. (**c**) Expression level of MOR in Vgat-Cre/ *Oprm1*$^{KI/KI}$ mice compared to wild-type mice determined by quantification of signals of MOR by immunostaining. The expression level of MOR was normalized to the mean gray value of MOR signal in each brain region of wild-type mice. (**d–f**) Effects of morphine (10 mg/kg, s.c.) on pain tests with tail

*Figure 4 continued on next page*

Figure 4 continued

immersion (50°C), hot plate (52°C) and von Frey tests in $Oprm1^{KI/KI}$ mice and Vgat-Cre/$Oprm1^{KI/KI}$ mice. n = 5–8 mice. Two-way ANOVA (d), $F_{10,143}$ = 63.70, p<0.0001; (e), $F_{7,88}$ = 0.4163, p=0.8898; (f), $F_{5,78}$ = 7.448, p<0.0001 with Bonferroni correction. (g) Time course of formalin-induced nocifensive behaviors in Vgat-Cre/$Oprm1^{KI/KI}$ mice compared to $Oprm1^{KI/KI}$ mice with subcutaneous morphine injection (10 mg/kg). n = 8 mice for each group. (h) Summary of the nocifensive behaviors in phase I (0–10 min) and phase II (10–60 min) of formalin test in Vgat-Cre/$Oprm1^{KI/KI}$ mice compared to $Oprm1^{KI/KI}$ mice with subcutaneous morphine injection (10 mg/kg). n = 8 mice for each group. Student's unpaired $t$-test, $t_{14}$ = 1.248, p=0.2324 (Phase I); $t_{14}$ = 1.388, p=0.1868 (Phase II). (i and j) Time-course effects of morphine (10 mg/kg, s.c.) on thermal (i) and mechanical (j) sensitivities on CFA-induced pain responses in $Oprm1^{KI/KI}$ mice (n = 8) and Vgat-Cre/$Oprm1^{KI/KI}$ mice (n = 10) on d7 and d14, respectively. Two-way ANOVA (i), $F_{8,144}$ = 1.515, p=0.1567; (j), $F_{8,144}$ = 0.5818, p=0.7917 with Bonferroni correction. (k and l) Thermal and mechanical pain tests during CFA-induced inflammatory pain in $Oprm1^{KI/KI}$ and Vgat-Cre/$Oprm1^{KI/KI}$ mice. n = 8 mice for each group. Two-way ANOVA (k), $F_{11,168}$ = 2.848, p=0.0019; (l), $F_{9,140}$ = 6.034, p<0.0001 with Bonferroni correction. (m) Time-course effects of β-FNA (10 mg/kg, i.p.) on mechanical sensitivity on CFA-induced pain responses in $Oprm1^{KI/KI}$ and Vgat-Cre/$Oprm1^{KI/KI}$ mice on d42 when the pain response was mostly recovered (n = 8). RM one-way ANOVA ($Oprm1^{KI/KI}$, $F_{9,63}$ = 0.6393, p=0.7592; Vgat-Cre/$Oprm1^{KI/KI}$, $F_{9,63}$ = 5.992, p<0.0001) with Bonferroni correction. (n) Schematic diagram for conditional knockout MOR from GABAergic neurons. Vgat-Cre mice were crossed with $Oprm1^{fl/fl}$ mice to generate Vgat-Cre/$Oprm1^{fl/fl}$ mice. (o and p) Thermal and mechanical pain tests during CFA-induced inflammatory pain in $Oprm1^{fl/fl}$ (n = 6) and Vgat-Cre/$Oprm1^{fl/fl}$ mice (n = 7). Two-way ANOVA (o), $F_{11,132}$ = 5.428, p<0.0001; (p), $F_{11,132}$ = 4.943, p<0.0001 with Bonferroni correction. Data are presented as mean ± SEM, **p<0.01, ***p<0.001.

The online version of this article includes the following source data and figure supplement(s) for figure 4:

**Source data 1.** Raw data of the behavioral tests and MOR expression level in Vgat-Cre/Oprm1-KI and Vgat-Cre/Oprm1-fl groups of mice.
**Figure supplement 1.** Expression pattern of the re-expressed MOR and functional role of MORs in GABAergic neurons in morphine analgesia.
**Figure supplement 2.** Functional role of MORs in GABAergic neurons in opioid analgesia.

which lost MORs selectively in GABAergic neurons (*Figure 4—figure supplement 2b*). Consistently, the effect of systemic morphine in tail immersion test was largely abolished in Vgat-Cre/$Oprm1^{fl/fl}$ mice, but similar analgesic effect was observed on hot plate and von Frey tests in Vgat-Cre/$Oprm1^{fl/fl}$ mice compared with the $Oprm1^{fl/fl}$ mice (*Figure 4—figure supplement 2c–f*). Thus, MORs in GABAergic neurons play diverse roles in mediating the effect of morphine in modulating acute pain.

We further examined the role of MORs in GABAergic neurons in mediating the effect of morphine in inflammatory pain. For the formalin-induced inflammatory pain, we found that after systemic saline injection, the licking and flinching behaviors induced by formalin was comparable between Vgat-Cre/$Oprm1^{KI/KI}$ and the $Oprm1^{KI/KI}$ littermates (*Figure 4—figure supplement 2g–h*). Systemic morphine application shift the time course of formalin-evoked nocifensive behaviors rightward in Vgat-Cre/$Oprm1^{KI/KI}$ mice compared to $Oprm1^{KI/KI}$ mice, whereas the total durations were comparable in both phase I and phase II between the two groups (*Figure 4g–h*). Furthermore, for CFA-induced chronic inflammatory pain, systemic morphine injection exhibited no significant effect on the responses evoked by thermal or mechanical stimuli, in either $Oprm1^{KI/KI}$ or Vgat-Cre/$Oprm1^{KI/KI}$ mice (*Figure 4i–j*), suggesting that the MORs in GABAergic neurons play insignificant role in mediating the analgesic effect of morphine in the CFA-induced inflammatory pain. In addition, the locomotion showed no difference between Vgat-Cre/$Oprm1^{KI/KI}$ mice and $Oprm1^{KI/KI}$ mice after morphine treatment (*Figure 4—figure supplement 2i–j*). Thus, MORs in GABAergic neurons are not involved in morphine analgesia under inflammatory pain conditions.

Next, we asked whether MORs in GABAergic neurons are involved in analgesia by endogenous opioids under chronic inflammatory pain condition. We injected CFA in the hindpaw of both Vgat-Cre/$Oprm1^{KI/KI}$ and $Oprm1^{KI/KI}$ mice, and found both groups of mice developed hyperalgesia from 2 hr to 7 days after CFA injection. Surprisingly, the Vgat-Cre/$Oprm1^{KI/KI}$ mice gradually recovered from hyperalgesia 3–4 weeks after CFA injection, but the $Oprm1^{KI/KI}$ mice showed persistent hyperalgesia even 7 weeks after CFA application (*Figure 4k–l*), indicating that MORs in GABAergic neurons play a critical role in mediating the analgesia by endogenous opioids. Next, we confirmed that the recovery was mediated by MORs with the pharmacological approach, and found that intraperitoneal injection of β-FNA, a specific MOR antagonist, reduced the mechanical threshold in Vgat-Cre/$Oprm1^{KI/KI}$ mice, but not in $Oprm1^{KI/KI}$ mice at 42 days after CFA injection (*Figure 4m*). Further, we confirmed the role of MORs in GABAergic neurons in mediating analgesia by endogenous opioid with the mice in which the MORs were selectively deleted in GABAergic neurons (*Figure 4n*). Consistently, we found that Vgat-Cre/$Oprm1^{fl/fl}$ mice exhibited significantly slower recovery from hyperalgesia induced by CFA, as compared to $Oprm1^{fl/fl}$ littermates (*Figure 4o–p*). Together, these results

indicate that the analgesia by endogenous opioids under chronic inflammatory pain condition is mainly mediated by MORs expressed in GABAergic neurons.

## MORs in the dorsal spinal cord mediate exogenous but not endogenous opioid analgesia

MORs are highly expressed in the dorsal spinal cord, and previous pharmacological studies indicate that these MORs play essential roles in opioid analgesia (*Kemp et al., 1996*; *Wang et al., 2018*). Given the potential non-specificity of pharmacological approaches, we further explored the functional role of MORs expressed in the dorsal spinal cord in exogenous and endogenous opioid analgesia with genetic approaches. We selectively expressed MORs in the dorsal spinal cord by crossing the *Oprm1*$^{KI/KI}$ mice with Lbx1-Cre mice (*Figure 5a*), in which Cre is expressed in the dorsal spinal cord (*Sieber et al., 2007*). Immunostaining results showed that MORs were rescued in the dorsal spinal cord but not DRG in Lbx1-Cre/*Oprm1*$^{KI/KI}$ compared to *Oprm1*$^{KI/KI}$ mice, and the MOR signal in the superficial layer of spinal cord was restricted in layers labeled with IB4 (*Figure 5b–c*). We next tested the functional roles of MORs in the dorsal spinal cord in opioid analgesia with these animals. Although the basal pain thresholds of Lbx1-Cre/*Oprm1*$^{KI/KI}$ and *Oprm1*$^{KI/KI}$ littermates were comparable (*Figure 5—figure supplement 1a–d*), we found morphine evoked significant analgesic effects in Lbx1-Cre/*Oprm1*$^{KI/KI}$ mice compared to the *Oprm1*$^{KI/KI}$ mice in tail immersion, hot plate and von Frey tests (*Figure 5d–f*). In formalin test, Lbx1-Cre/*Oprm1*$^{KI/KI}$ and *Oprm1*$^{KI/KI}$ mice exhibited similar nocifensive behaviors (*Figure 5—figure supplement 1e–f*). Systemic injection of morphine produced significant antinociceptive effect in the early period, but not in the phase II in Lbx1-Cre/*Oprm1*$^{KI/KI}$ mice compared with *Oprm1*$^{KI/KI}$ mice (*Figure 5g–h*). In CFA-induced chronic inflammatory pain, morphine only exerted antinociceptive effect with Lbx1-Cre/*Oprm1*$^{KI/KI}$ mice in thermal but not mechanical pain test (*Figure 5i–j*). We further determined the functional role of MORs expressed in dorsal spinal cord in endogenous opioid analgesia. In chronic inflammatory pain induced by CFA, neither Lbx1-Cre/*Oprm1*$^{KI/KI}$ nor *Oprm1*$^{KI/KI}$ mice recovered from hyperalgesia even 7 weeks later (*Figure 5k–l*), indicating that MORs in the dorsal spinal cord play an insignificant role in endogenous opioid analgesia.

We next confirmed the functional role of MORs expressed in dorsal spinal cord using conditional knockout mice (*Figure 5—figure supplement 1g*), which lost the MORs in the dorsal spinal cord but not DRG (*Figure 5—figure supplement 1h*). We found that the analgesic effects of morphine in tail immersion and hot plate tests but not von Frey test were significantly reduced in Lbx1-Cre/*Oprm1*$^{fl/fl}$ mice compared to *Oprm1*$^{fl/fl}$ mice (*Figure 5—figure supplement 1i–l*). In formalin test, Lbx1-Cre/*Oprm1*$^{fl/fl}$ and *Oprm1*$^{fl/fl}$ mice showed similar nocifensive behaviors (*Figure 5—figure supplement 1m*). The analgesic effect of morphine was reduced in the phase I of formalin test in Lbx1-Cre/*Oprm1*$^{fl/fl}$ mice, but was intact in phase II (*Figure 5—figure supplement 1m*). In CFA-induced inflammatory pain, Lbx1-Cre/*Oprm1*$^{fl/fl}$ and *Oprm1*$^{fl/fl}$ mice were indistinguishable in recovery from hyperalgesia (*Figure 5—figure supplement 1n–o*), further confirming that MORs expressed in the dorsal spinal cord play an insignificant role in endogenous opioid analgesia. Taken together, our results suggest that MORs expressed in the dorsal spinal cord are partially involved in the morphine analgesia but not endogenous opioid analgesia.

## Functional roles of MORs expressed in different types of neurons at the spinal level

MORs are likely expressed in different sub-populations of spinal neurons (*Wang et al., 2018*). We further determined the distribution of MOR$^+$ neurons in the spinal cord using *in situ* hybridization. Consistent with previous observations (*Wang et al., 2018*), we found that *Oprm1*$^+$ neurons are widely distributed in the spinal cord, and that a significant proportion of the *Oprm1*$^+$ neurons localized in superficial layers of dorsal spinal cord (*Figure 6a–b*). We next determined the identity of MOR$^+$ neurons by using *in situ* hybridization, and found that MORs are expressed in both GABAergic and glutamatergic dorsal spinal neurons with comparable percentage (*Figure 6c–d*).

We further determined the functional role of MORs expressed in excitatory neurons at the spinal level by using the Vglut2-Cre/*Oprm1*$^{KI/KI}$ mice (*Figure 3a*). MORs were re-expressed in both spinal cord and DRG neurons in Vglut2-Cre/*Oprm1*$^{KI/KI}$ mice (*Figure 6e–f*). We selectively activated MORs expressed in the glutamatergic neurons at the spinal level by intrathecal injecting of morphine, and

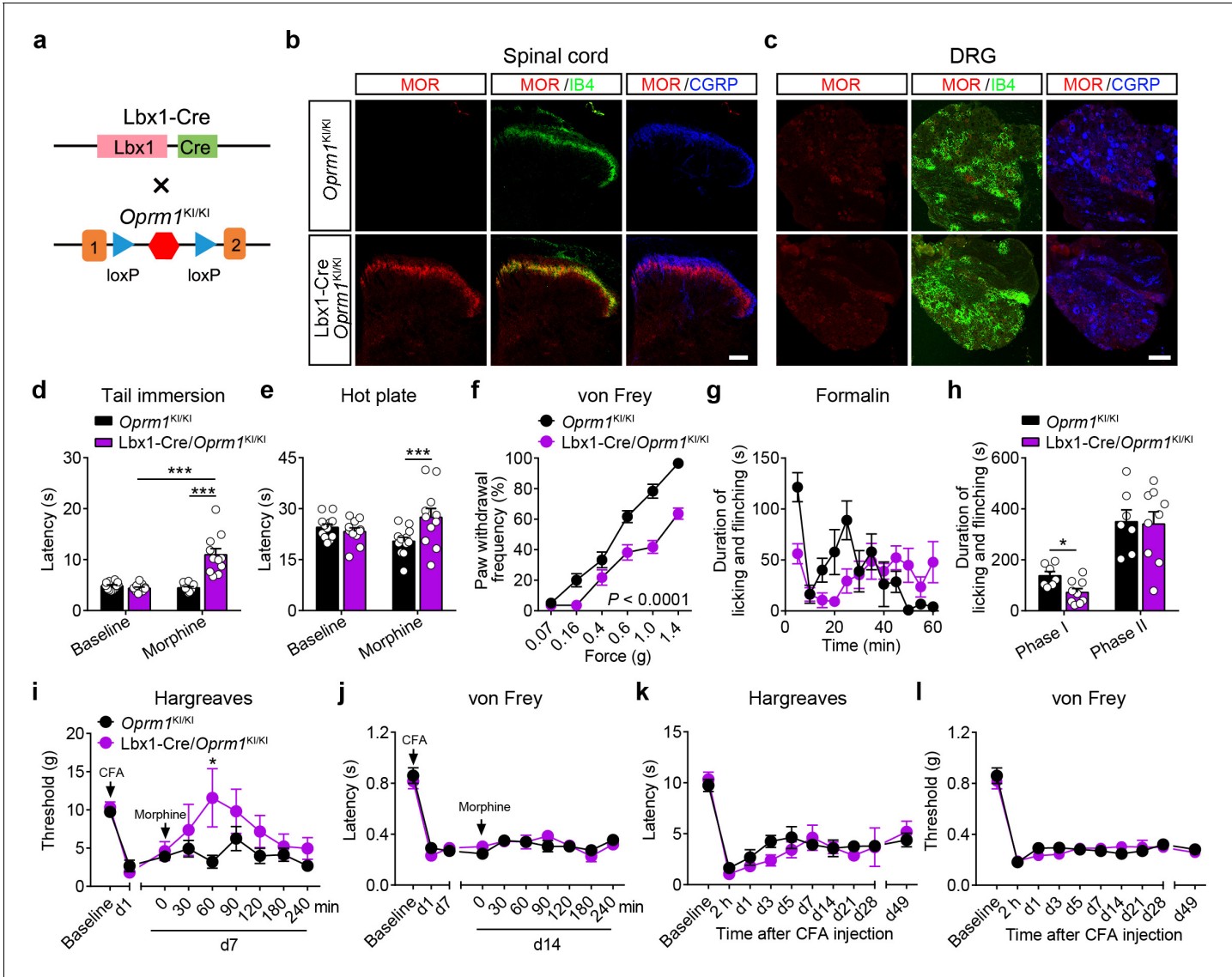

**Figure 5.** MORs in dorsal spinal cord mediate morphine analgesia. (a) Schematic diagram for re-expression of MORs in dorsal spinal neurons. Lbx1-Cre mice were crossed with *Oprm1*^KI/KI^ mice to generate Lbx1-Cre/*Oprm1*^KI/KI^ mice. Red hexagon: Stop cassette. (b and c) Co-immunostaining of MOR with IB4 and CGRP in *Oprm1*^KI/KI^ and Lbx1-Cre/*Oprm1*^KI/KI^ spinal cord (left) and DRG (right). Scale bars, 100 μm. (d–f) Effects of morphine (10 mg/kg, s. c.) on pain tests with tail immersion (50°C), hot plate (52°C) and von Frey tests in *Oprm1*^KI/KI^ (n = 12) and Lbx1-Cre/*Oprm1*^KI/KI^ (n = 11) mice. Two-way ANOVA (d), $F_{1,42}$ = 33.80, p<0.0001; (e), $F_{1,42}$ = 7.038, p=0.0112; (f), $F_{5,126}$ = 5.717, p<0.0001 with Bonferroni correction. (g) Time course of formalin-induced nocifensive behaviors in Lbx1-Cre/*Oprm1*^KI/KI^ (n = 9) mice compared with *Oprm1*^KI/KI^ (n = 7) littermates with subcutaneous morphine injection (10 mg/kg). (h) Summary of the nocifensive behaviors in phase I (0–10 min) and phase II (10–60 min) of formalin test in Lbx1-Cre/*Oprm1*^KI/KI^ (n = 9) mice compared with *Oprm1*^KI/KI^ (n = 7) littermates with subcutaneous morphine injection (10 mg/kg). Student's unpaired *t* test, $t_{14}$ = 2.924, p=0.0111 (Phase I); $t_{14}$ = 0.1382, p=0.8921 (Phase II). (i and j) Time-course effects of morphine (10 mg/kg, s.c.) on thermal (i) and mechanical (j) sensitivities on CFA-induced pain responses in *Oprm1*^KI/KI^ mice (n = 8) and Lbx1-Cre/*Oprm1*^KI/KI^ mice (n = 5) on d7 and d14, respectively. Student's unpaired *t*-test, $t_{11}$ = 2.681, p=0.0214 (i), 60 min. (k and l) Thermal and mechanical pain tests during CFA-induced inflammatory pain in *Oprm1*^KI/KI^ (n = 8) and Lbx1-Cre/*Oprm1*^KI/KI^ (n = 5) mice. Two-way ANOVA (k), $F_{9,110}$ = 0.6637, p=0.7400; (l), $F_{9,110}$ = 0.6157, p=0.7814 with Bonferroni correction. Data are presented as mean ± SEM, *p<0.05, ***p<0.001.

The online version of this article includes the following source data and figure supplement(s) for figure 5:

**Source data 1.** Raw data of the behavioral tests in Lbx1-Cre/Oprm1-KI and Lbx1-Cre/Oprm1-fl groups of mice.
**Figure supplement 1.** Functional role of MORs in dorsal spinal neurons in opioid analgesia.

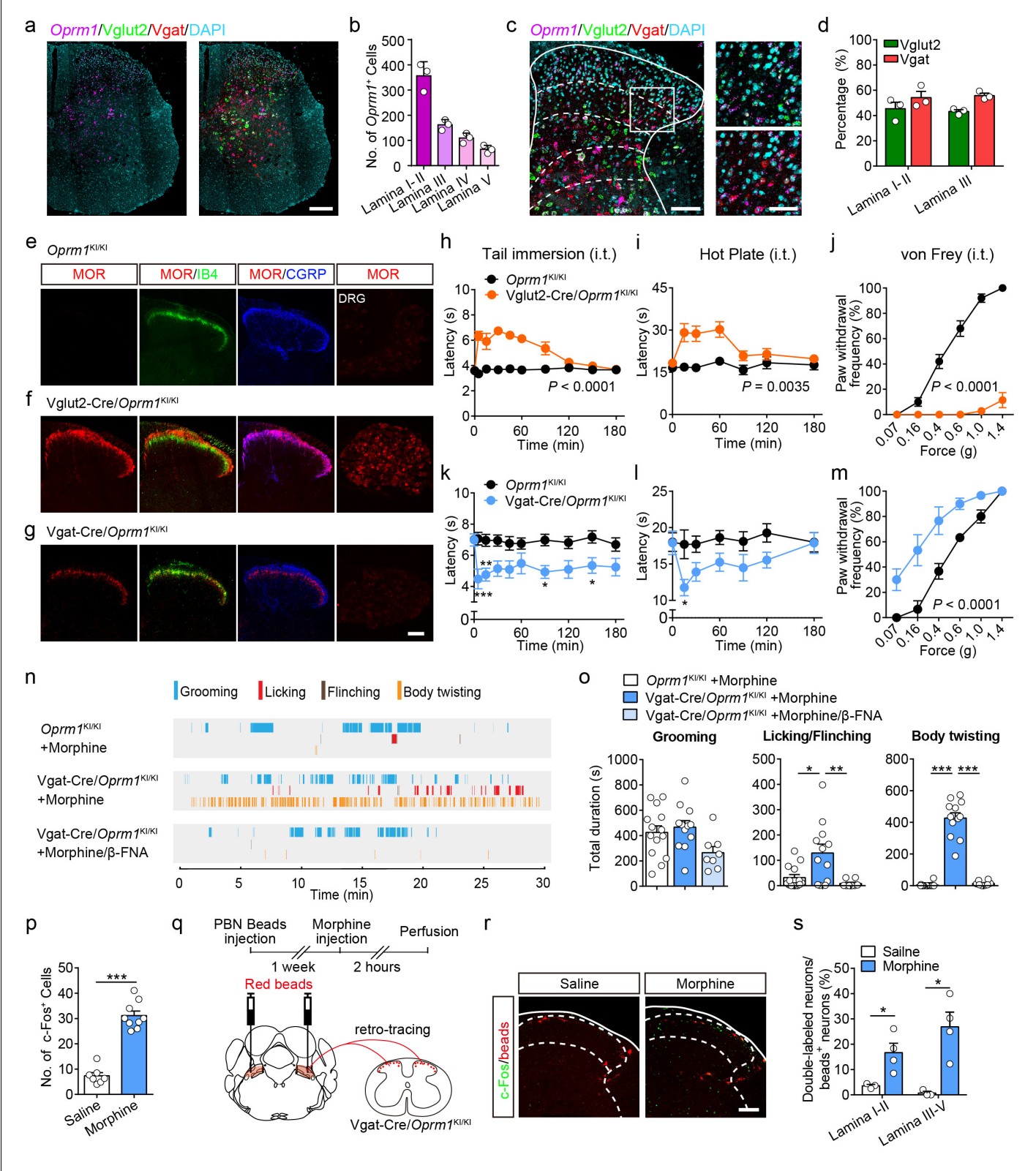

**Figure 6.** Activation of MORs expressed in spinal GABAergic neurons induced hyperalgesia. (a) Expression of *Oprm1*, Vglut2 and Vgat mRNA in the spinal cord of wild-type mice. Scale bar, 200 μm. (b) Quantification of *Oprm1*+ cells in the dorsal spinal cord. (c) Expression of *Oprm1*, Vglut2 and Vgat mRNA in the dorsal spinal cord of wild-type mice. Scale bars, 100 μm (left), 50 μm (right). (d) Percentage of Vglut2+ and Vgat+ neurons in *Oprm1*+ neurons in the dorsal spinal cord. n = 3 mice. (e–g) Co-immunostaining of MOR, IB4 and CGRP in dorsal spinal cord and immunostaining of MOR in

*Figure 6 continued on next page*

Figure 6 continued

DRG of *Oprm1*^KI/KI, Vglut2-Cre/*Oprm1*^KI/KI and Vgat-Cre/*Oprm1*^KI/KI mice. Scale bar, 100 μm. (h–j) Effect of morphine (1.0 nmol/ 5 μl, intrathecal, i.t.) on pain tests with tail immersion (50°C), hot plate (52°C) and von Frey tests in Vglut2-Cre/*Oprm1*^KI/KI mice compared to *Oprm1*^KI/KI littermates. n = 7–10 mice. Two-way ANOVA (h), $F_{9,130}$ = 12.47, p<0.0001; (i), $F_{6,91}$ = 7.038, p=0.0035; (j), $F_{5,90}$ = 56.15, p<0.0001 with Bonferroni correction. (k–m) Effect of morphine (1 nmol/ 5 μl, i.t.) on pain tests with tail immersion (48°C), hot plate (52°C) and von Frey tests in Vgat-Cre/*Oprm1*^KI/KI mice compared to *Oprm1*^KI/KI littermates. n = 6–9 mice. Two-way ANOVA (k), $F_{9,150}$ = 1.162, p=0.3232; (l), $F_{6,98}$ = 1.206, p=0.3097; (m), $F_{5,60}$ = 3.405, p=0.0090 with Bonferroni correction. (n) Representative raster plots illustrating grooming, licking, flinching and body twisting behaviors in *Oprm1*^KI/KI (top), Vgat-Cre/*Oprm1*^KI/KI (middle) mice following morphine injection (1 nmol/ 5 μl, i.t.) and Vgat-Cre/*Oprm1*^KI/KI (bottom) mice following intrathecal injection of mixture of morphine (1 nmol/ 5 μl) and β-FNA (10 nmol/ 5 μl). (o) Grooming and nocifensive behaviors induced by intrathecal morphine injection in Vgat-Cre/*Oprm1*^KI/KI and *Oprm1*^KI/KI mice. n = 8–14 mice. One-way ANOVA (left to right: $F_{2,31}$ = 3.333, p=0.0488; $F_{2,31}$ = 7.350, p=0.0024; $F_{2,31}$ = 137.8, p<0.0001) with Bonferroni correction. (p) Number of morphine- (1 nmol/ 5 μl, i.t., n = 9) or saline-induced (n = 7) c-Fos+ neurons in dorsal spinal cord of Vgat-Cre/*Oprm1*^KI/KI mice. Student's unpaired *t*-test, $t_{14}$ = 10.21, p<0.0001. (q) Schematic diagram for experimental timeline. (r) Representative images of c-Fos+ and beads+ neurons in dorsal spinal cord of Vgat-Cre/*Oprm1*^KI/KI mice after saline or morphine (1 nmol/ 5 μl, i.t.) administration. Scale bar, 100 μm. (s) Percentage of beads+ neurons in c-Fos+ neurons in the dorsal spinal cord. n = 3–4 mice. Student's unpaired *t*-test, $t_5$ = 2.954, p=0.0317; $t_5$ = 3.813, p=0.0125. Data are presented as mean ± SEM; *p<0.05, **p<0.01, ***p<0.001.

The online version of this article includes the following source data and figure supplement(s) for figure 6:

**Source data 1.** Raw data of the behavioral tests in Vglut2-Cre/Oprm1-KI and Vgat-Cre/Oprm1-KI groups of mice with intrathecal morphine injection.
**Figure supplement 1.** Functional role of MORs in primary neurons in opioid analgesia.
**Figure supplement 1—source data 1.** Raw data of the behavioral tests in SNS-Cre/Oprm1-KI mice.

found morphine induced analgesic effect on both thermal and mechanical pain tests in Vglut2-Cre/*Oprm1*^KI/KI mice but not *Oprm1*^KI/KI mice (***Figure 6h–j***). Since MORs in the DRG neurons are also located in the central terminals (***Scherrer et al., 2009***), the analgesic effect of intrathecal injection of morphine is also likely mediated by targeting at the MORs originating from DRG. To examine this possibility, we selectively expressed MORs in small DRG neurons with a DRG specific Cre mouse line (***Agarwal et al., 2004***), and found that MOR immunoreactivity was restricted to CGRP-positive lamina I and II outer layers in the dorsal spinal cord of SNS-Cre/*Oprm1*^KI/KI mice (***Figure 6—figure supplement 1a–b***). In contrast, selective activation of MORs in the DRG neurons by using SNS-Cre/*Oprm1*^KI/KI mice had no significant effect on the acute pain or formalin-induced nocifensive behaviors, and only slightly suppressed thermal pain in CFA-induced inflammatory pain (***Figure 6—figure supplement 1c–k***). Taken together, our data indicate that MORs expressed in spinal glutamatergic neurons are the major target for morphine analgesia at the spinal level.

We next determined the functional role of MORs in GABAergic neurons at the spinal level using Vgat-Cre/*Oprm1*^KI/KI mice, in which expression of MORs was rescued in GABAergic neurons. We found that MORs were detected in the spinal lamina II inner layer labeled by IB4 and deeper layers (***Figure 6g***). Selective activation of MORs expressed in spinal GABAergic neurons with Vgat-Cre/*Oprm1*^KI/KI mice decreased both thermal sensitivity and mechanical thresholds in acute pain tests (***Figure 6k–m***). This manipulation also induced robust nocifensive responses including hindpaw licking, hindpaw flinching and body twisting behaviors, which could be blocked by a specific MOR antagonist, β-FNA (***Figure 6n–o***, ***Video 1***). Consistent with the behavioral results, we found that activation of MORs expressed in spinal GABAergic neurons by intrathecal injection of morphine also significantly increased the number of neurons expressing c-Fos, a neuronal activity marker (***Figure 6p***). Furthermore, we determined whether the activity of projection neurons was also increased after intrathecal morphine injection in Vgat-Cre/*Oprm1*^KI/KI mice. About 80% lamina I projection neurons project to PBN (***Todd, 2010***), and the spino-parabrachial pathway has been implicated in the processing of nociceptive signal (***Han et al., 2015***). We thus labeled the spinal projection neurons by

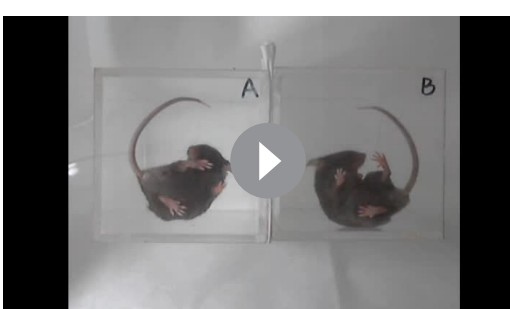

**Video 1.** Intrathecal morphine-induced nocifensive behaviors in Vgat-Cre/*Oprm1*^KI/KI mice. Left, *Oprm1*^KI/KI mice. Right, Vgat-Cre/*Oprm1*^KI/KI mice. Morphine (1.0 nmol/ 5 μl) was intrathecally injected.
https://elifesciences.org/articles/55289#video1

injection of retro-beads in the PBN (*Figure 6q*). We found that intrathecal morphine administration significantly increased the percentage of c-Fos$^+$/beads$^+$ neurons in both superficial and deeper dorsal horn of Vgat-Cre/*Oprm1*$^{KI/KI}$ mice (*Figure 6r–s*), suggesting that the PBN-projecting neurons were excited. These results further support the notion that activation of MORs in spinal GABAergic neurons induces hyperalgesia.

## MORs in the PBN mediate morphine analgesia in inflammatory pain

Our results indicate that the MORs expressed in the glutamatergic neurons in the brain play a crucial role in morphine analgesia but not endogenous opioid analgesia. We further tested this hypothesis. Given that MORs are highly expressed in glutamatergic neurons in external lateral part of the PBN (*Figure 1d–e*) and that the PBN is involved in pain processing (*Barik et al., 2018*; *Han et al., 2015*; *Huang et al., 2019*; *Rodriguez et al., 2017*), we thus examined the role of MORs expressed in the PBN in opioid analgesia. We generated a MOR-iCreER$^{T2}$ mouse line to label the MOR$^+$ neurons, and performed double *in situ* hybridization in MOR-iCreER$^{T2}$ × Ai9 mice to verify the efficiency and specificity of the MOR-iCreER$^{T2}$ mouse line (*Figure 7—figure supplement 1a–b*). We found that 94.5 ± 1.8% *tdTomato*$^+$ neurons were positive for *Oprm1*, indicating high specificity, whereas the labeling efficiency varied among different brain areas, on average 46.8 ± 2.8% (range: 15.3 ± 3.1% to 85.6 ± 0.5%) of *Oprm1*$^+$ neurons were positive for *tdTomato* (*Figure 7—figure supplement 1c–f*). The labeling efficiency of MOR$^+$ neurons in PBN was 71.3 ± 5.2%.

Next, we examined the response of MOR$^+$ neurons in the PBN to noxious stimuli by fiber photometry measurements of fluorescence signals of a calcium indicator GCaMP6s, which was specifically expressed in PBN MOR$^+$ neurons (*Figure 7a*). We observed elevated GCaMP6s fluorescent signals in the PBN during noxious mechanical stimuli, including tail and hindpaw pinch (*Figure 7b*). For the thermal stimuli, high temperature (52°C) but not the warm temperature (40°C) stimuli significantly increased the fluorescent signals of GCaMP6s (*Figure 7c*). The quantitative analysis of the fluorescent signal following stimuli onset showed that MOR$^+$ neurons in the PBN were activated by both noxious mechanical and thermal stimuli (*Figure 7d*). The increase of fluorescent signal during noxious stimulation was not due to movement artifact, because the change of fluorescent signal of EYFP was not significant (*Figure 7d*). Moreover, systemic morphine application significantly decreased the calcium response of MOR$^+$ neurons to tail pinch, and this effect largely diminished 5 hr after morphine injection (*Figure 7e–g*). These data supporting the idea that MORs expressed in the PBN are involved in modulating nociceptive information processing.

Next, we tested whether the MORs expressed in the PBN mediate morphine analgesia. We re-expressed MORs in PBN neurons by injecting an AAV expressing Cre-EGFP (AAV-hSyn-Cre-EGFP) bilaterally in the PBN of *Oprm1*$^{KI/KI}$ mice (*Figure 7h*), and AAV-hSyn-EGFP was used as control. These mice are referred to as PBN-Cre/*Oprm1*$^{KI/KI}$ and PBN-EGFP/*Oprm1*$^{KI/KI}$ mice hereafter, respectively. We found that MOR expression in the PBN was largely restored after injection of AAV-Cre virus (*Figure 7i*), and the expression pattern was comparable to that in the wild-type mice (*Figure 2—figure supplement 1a*). Behavioral tests showed that there was no significant difference in basal pain thresholds or the locomotor activity between the two groups (*Figure 7—figure supplement 2a–f*). We next tested the analgesic effect of morphine on acute pain, and found that systemic morphine injection did not significantly alter the nociceptive responses to thermal or mechanical stimulation in either PBN-Cre/*Oprm1*$^{KI/KI}$ or PBN-EGFP/*Oprm1*$^{KI/KI}$ mice (*Figure 7—figure supplement 2b–d*). In addition, systemic injection of morphine had no significant effect on the locomotion of PBN-Cre/*Oprm1*$^{KI/KI}$ and PBN-EGFP/*Oprm1*$^{KI/KI}$ mice (*Figure 7—figure supplement 2e–f*). These results suggest that MORs expressed in PBN are not involved in the analgesic effect of morphine in acute pain.

We further inquired whether MORs expressed in the PBN modulate inflammatory pain. We examined the effect of morphine on formalin-induced acute inflammatory pain in PBN-Cre/*Oprm1*$^{KI/KI}$ and PBN-EGFP/*Oprm1*$^{KI/KI}$ mice. These two groups of mice showed comparable nocifensive behavior in the formalin test (*Figure 7—figure supplement 2g–h*), while systemic morphine injection significantly reduced the nocifensive behavior in both phase I and phase II in PBN-Cre/*Oprm1*$^{KI/KI}$ mice in comparison to PBN-EGFP/*Oprm1*$^{KI/KI}$ mice (*Figure 7j–k*), indicating an important role of MORs in the PBN in morphine analgesia in acute inflammatory pain. We next explored the role of PBN MORs in CFA-induced chronic inflammatory pain. The mice in both groups showed hyperalgesia after CFA application, while systemic morphine injection significantly elevated the thermal and mechanical

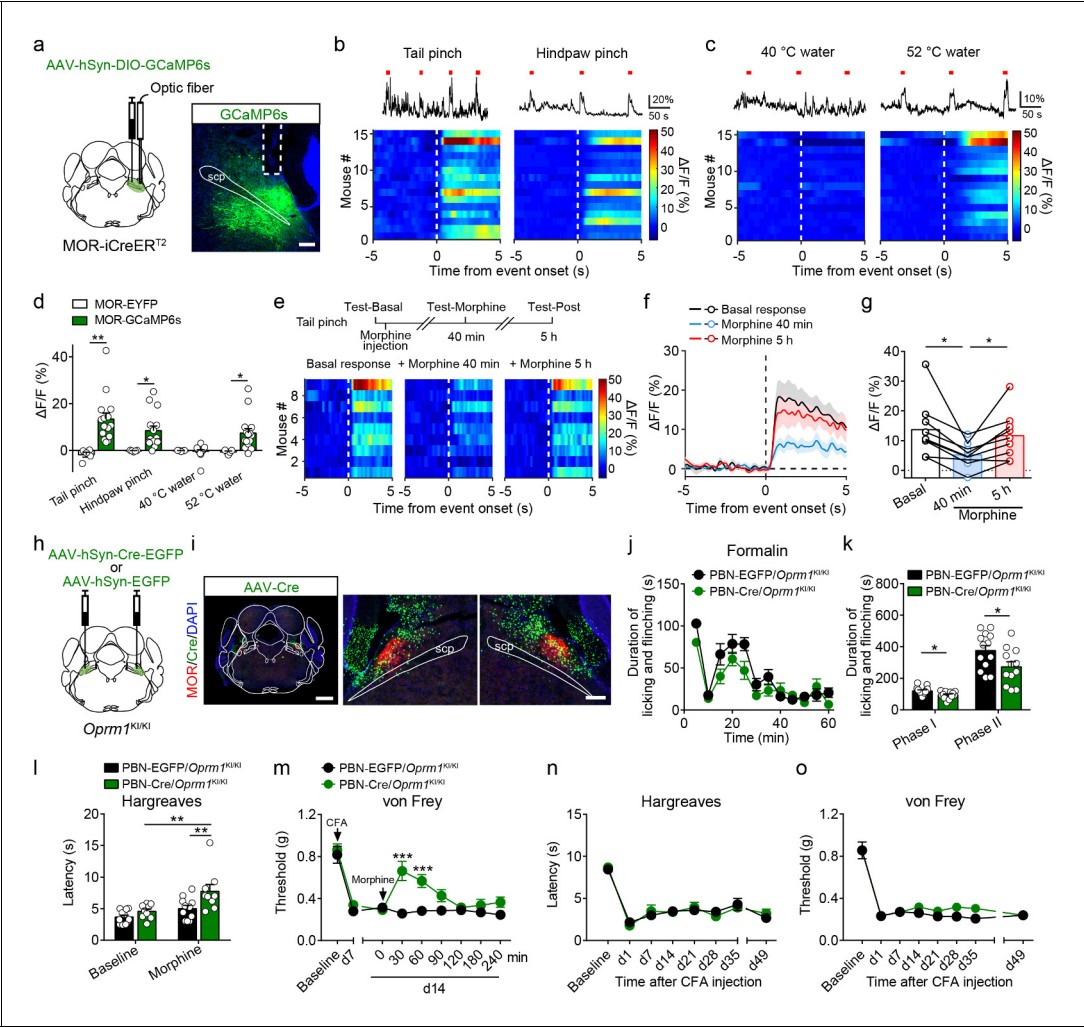

**Figure 7.** Functional role of MORs expressed in the PBN in opioid analgesia. (a) Schematic diagram showing stereotaxic injection into PBN and representative expression of AAV-hSyn-DIO-GCaMP6s virus in MOR-iCreER[T2] mice. Dashed line outlines the track of optical fiber. Scale bar, 200 μm. (b) Representative photometry traces (top) and averaged GCaMP6s fluorescence dynamics relative to the pinch onset (bottom) in response to tail (left) and hindpaw (right) pinch. Each red bar represents a pinch event. (c) Representative photometry traces (top) and averaged GCaMP6s fluorescence dynamics relative to the onset (bottom) in response to 40°C (left) and 52°C (right) tail immersion. Each red bar represents an event. (d) Comparison of the averaged fluorescence signal change between EYFP (n = 5) and GCaMP6s (n = 15) group during onset period (0–5 s) for each stimulation. Student's unpaired $t$-test, tail pinch: $t_{18}$ = 3.259, p=0.0044; hindpaw pinch: $t_{18}$ = 2.435, p=0.0255; 40°C hot water: $t_{18}$ = 0.1860, p=0.8545; 52°C hot water: $t_{18}$ = 2.386, p=0.0293. (e) Diagram for experiments and individual response of MOR[+] neurons in PBN during tail pinch in each test. (f) Time course of calcium activity in response to tail pinch in each session. (g) Comparison of the averaged fluorescence signal change during tail pinch onset period (0–5 s) in each session. n = 9 mice. One-way ANOVA ($F_{2,16}$ = 11.46, p=0.0008) with Bonferroni correction. (h) Schematic diagram of bilateral PBN stereotaxic injection of AAV-hSyn-Cre-EGFP or AAV-hSyn-EGFP virus. (i) Representative MOR re-expression in *Oprm1*[KI/KI] mice with bilateral AAV-hSyn-Cre-EGFP virus in PBN. Scale bars, 1 mm (left), 200 μm (middle and right). (j) Time course of formalin-induced nocifensive behaviors in control mice (n = 13) and PBN-Cre/*Oprm1*[KI/KI] mice (n = 12) with subcutaneous morphine injection (10 mg/kg). (k) Summary of the nocifensive behaviors in phase I (0–10 min) and phase II (10–60 min) of formalin test in control mice (n = 13) and PBN-Cre/*Oprm1*[KI/KI] mice (n = 12) with subcutaneous morphine injection (10 mg/kg). Student's unpaired $t$-test, $t_{23}$ = 2.682, p=0.0133 (Phase I); $t_{23}$ = 2.253, p=0.0341 (Phase II). (l and m) Effects of morphine (10 mg/kg, s.c.) on thermal sensitivity at d7 (l) and mechanical sensitivity at d14 (m) on CFA-induced pain in control mice (n = 11) and PBN-Cre/*Oprm1*[KI/KI] mice (n = 9). Two-way ANOVA (l), $F_{1,36}$ = 2.222, p=0.1448; (m), $F_{8,162}$ = 4.550, p<0.0001 with Bonferroni correction. (n and o) Pain tests for thermal and mechanical sensitivities during CFA-induced inflammatory pain in PBN-EGFP/*Oprm1*[KI/KI] and PBN-Cre/*Oprm1*[KI/KI] mice. n = 7 mice for each group. Two-way ANOVA (n), $F_{7,96}$ = 0.3247, p=0.9411; (o), $F_{7,96}$ = 0.8869, p=0.5200 with Bonferroni correction. Data are presented as mean ± SEM; *p<0.05, **p<0.01, ***p<0.001.

The online version of this article includes the following source data and figure supplement(s) for figure 7:

**Source data 1.** Raw data of the fiberphotometry recording in Oprm1-GCaMP6s mice and behavioral tests in PBN-Cre/Oprm1-KI mice.

**Figure supplement 1.** Generation and verification of MOR-iCreER[T2] mice.

**Figure supplement 2.** Functional role of MORs expressed in the PBN in morphine analgesia.

thresholds in the PBN-Cre/*Oprm1*[KI/KI] mice but not the PBN-EGFP/*Oprm1*[KI/KI] mice (*Figure 7l–m*). However, both groups of mice exhibited comparable persistent pain and did not recover from hyperalgesia in long-term test after CFA application (*Figure 7n–o*), indicating that MORs expressed in the PBN are not involved in endogenous opioid analgesia. Taken together, our results indicate that the MORs expressed in PBN glutamatergic neurons play an important role in mediating morphine analgesia in both acute and chronic inflammatory pain, but are not involved in endogenous opioid analgesia.

## Discussion

Our study revealed that MORs expressed in glutamatergic and GABAergic neurons play diverse roles in mediating opioid analgesia. We found that analgesia by systemic morphine is largely mediated by MORs expressed in Vglut2[+] glutamatergic neurons. By contrast, the MORs expressed in GABAergic neurons are crucial for analgesia by endogenous opioids during chronic inflammatory pain. In addition, MORs expressed at the spinal level and in the PBN play an important role in the analgesic effect of morphine in acute pain and inflammatory pain, respectively, but not in endogenous opioid analgesia. These results demonstrated that analgesia by exogenous and endogenous opioids are mediated by MORs expressed in different neuronal populations.

### Distribution of MORs in different neuronal populations

We determined the identity of the *Oprm1*[+] neurons with triple FISH. We found that *Oprm1* was widely expressed in different population of neurons, with the percentage of *Oprm1* in different cell types varying across different brain areas and the spinal cord. For example, in the PAG, although it was thought that MOR is highly expressed in GABAergic neurons and that activation of MOR inhibits GABAergic synaptic transmission (*Vaughan et al., 1997*), we found that a significant proportion of *Oprm1*[+] neurons were glutamatergic. In addition, we found that MORs expressed in both Vglut2[+] excitatory neurons and GABAergic neurons in the dorsal spinal cord with comparable proportion, which is in contrast to a previous study in rat showing that MORs were mainly expressed in excitatory neurons in the spinal cord (*Kemp et al., 1996*). Complementing to the above results, we also revealed the distribution of MORs expressed in different types of neurons with immunohistochemistry using genetically modified mice (*Figures 3b* and *4b*; *Figure 3—figure supplement 2b*; *Figure 4—figure supplement 2b*). This is important as MORs are highly expressed in presynaptic terminals and play important roles in gating the synaptic transmission presynaptically (*Li et al., 2016*; *Vaughan et al., 1997*). Together, these results provide a comprehensive view of MOR distribution in the central nervous system.

### Neural mechanisms underlying morphine analgesia

Our results identified the neuronal population that is responsible for analgesia by systemic morphine. Although early studies have implicated MORs expressed in GABAergic neurons in opioid analgesia (*Fields, 2004*), the key neuronal population underlying the analgesic effect of systemic morphine is not known. Our study demonstrated that analgesia by systemic morphine is mainly mediated by MORs expressed in Vglut2[+] glutamatergic neurons. We confirmed these results with different genetic approaches. Our results are in line with recent genetic studies, showing that MORs expressed in forebrain GABAergic neurons were not involved in morphine analgesia (*Charbogne et al., 2017*; *Cui et al., 2014*). However, previous pharmacological studies showed that local injection of morphine in some forebrain and midbrain areas enriched with GABAergic neurons produced analgesia (*Cohen et al., 1984*; *Manning et al., 1994*; *Yaksh and Rudy, 1977*). This could be explained by the possibility that local injection of morphine could activate not only MORs expressed in local neurons but also MORs expressed in presynaptic terminals that originate from other brain areas, including those with abundant glutamatergic neurons. This emphasizes the importance of dissecting the mechanism underlying morphine analgesia by combining genetic and pharmacological approaches. Glutamatergic neurons consist of several sub-populations, which could be marked by Vglut1, Vglut2 and Vglut3. Given that Vglut2[+] glutamatergic neurons represent the largest population, here we only examined the functional role of MORs expressed in Vglut2[+] glutamatergic neurons in morphine analgesia. It is likely that other sub-populations of glutamatergic neurons

might also involve in analgesic effect of opioids, given MORs are also expressed in the Vglut1[+] neurons.

MORs are widely expressed in the brain and at the spinal level. We found that MORs at different areas play diverse roles in morphine analgesia in acute and inflammatory pain. At the spinal level, MORs are expressed in the DRG and spinal cord (*Scherrer et al., 2009*; *Wang et al., 2018*). Our results indicate that MORs expressed in the spinal cord are important for morphine analgesia in acute pain, and are partially involved in analgesia during inflammatory pain. This is evidenced by the data showing that selective activation of MORs in the dorsal spinal cord evoked analgesic effect in phase I, but not the phase II in formalin test. Our results are different from that obtained in rats by ablation of MOR-expressing neurons in the spinal cord, in which the effect of morphine on phase II was also decreased (*Kline and Wiley, 2008*). This analgesic effect of morphine in the spinal cord is most likely mediated by MORs expressed in the glutamatergic neurons, as activation of MORs in spinal GABAergic neurons induced allodynia and hyperalgesia. Additionally, MORs in the DRG is only partially involved in morphine analgesia in thermal hyperalgesia during CFA-induced chronic pain, which is likely due to upregulation of MORs during inflammation (*Ballet et al., 2003*). A recent study showed that Vgat is expressed in primary sensory neurons (*Du et al., 2017*). However, we observed no significant MOR signal in the DRG of Vgat-Cre/*Oprm1*[KI/KI] mice, suggesting that MORs are barely expressed in Vgat[+] DRG neurons. Our behavioral data are consistent with previous studies with mice lacking MOR in the DRG (*Corder et al., 2017*; *Weibel et al., 2013*). Together with the data obtained with Vglut2-Cre/*Oprm1*[KI/KI] mice (*Figure 3*), these data indicate that the MORs expressed at the supraspinal level play a more important role in mediating the analgesic effect of morphine in chronic pain. This is further evidenced by the observation that selective activation of MORs in the PBN, which consists of mostly glutamatergic neurons, significantly reduced inflammatory pain but not acute pain. However, the PBN is most likely one of many key brain areas involved in analgesia by systemic morphine, as re-expressing MOR in the PBN only partially rescued the effect of systemic morphine.

Pain is a complex feeling that consists of both sensory and affective components. In addition to suppressing pain perception, opioid analgesics also play an important role in relieving the emotional discomfort of pain (*Corder et al., 2018*; *Price et al., 1985*). For example, activation of MORs in the anterior cingulate cortex significantly alleviated the aversion caused by chronic pain (*LaGraize et al., 2006*). In the present study, we only examined the analgesic effect of morphine for the sensory component of pain. Further studies are warranted to examine the functional role of MORs expressed in different population of neurons in mediating the opioid analgesics in affective component of pain.

## Analgesia by endogenous opioids in chronic inflammatory pain

Previous pharmacological studies suggest that endogenous opioids are involved in both acute and chronic pain modulation (*Corder et al., 2013*), although several genetic studies did not reach consensus about the role of MORs in endogenous pain control (*Maldonado et al., 2018*). Our study revealed that MORs expressed in GABAergic neurons, but not glutamatergic neurons, play a critical role in analgesia by endogenous opioids. This is evidenced by the data showing that mice with selective expression of MORs in GABAergic, but not Vglut2[+] glutamatergic neurons, recovered from CFA-induced hyperalgesia, while animals that lacking MORs did not. This is in line with previous thought that activation of MOR in GABAergic neurons lead to pain suppression (*Al-Hasani and Bruchas, 2011*; *Fields, 2004*). Furthermore, our results indicate that MORs expressed in the brain are responsible for endogenous opioid analgesia observed in chronic inflammatory pain, as conditional knockout of MOR from the dorsal spinal cord did not affect endogenous opioid analgesia. However, our results are inconsistent with a previous study indicating that constitutive MOR activity in the spinal cord produces endogenous opioid analgesia (*Corder et al., 2013*). Given that MORs could be expressed presynaptically in descending fibers and that activation of supraspinal MORs leads to an increase of descending inhibitory component (*Gogas et al., 1991*), it remains possible that the antinociceptive effect of endogenous opioid in the spinal cord relies on the activation of MORs expressed presynaptically in the descending pathway. Future studies on the specific neuronal populations that mediate endogenous opioid analgesia are needed.

An unexpected finding in the present study is that the endogenous opioid analgesia is mediated by mechanism different from exogenous opioid analgesia. Enhancement of endogenous opioid peptide release and opioid-independent constitutive activation of MOR have both been implicated in

endogenous opioid analgesia (*Corder et al., 2018*; *Maldonado et al., 2018*; *Stein, 2016*). Thus, one possibility is that the endogenous opioid peptides released during inflammation targeted at MORs expressed in subpopulation of neurons other than those MORs activated by morphine. Consistently, several brain areas including amygdala, in which MORs are mostly expressed in GABAergic neurons, were shown to be activated during sustained muscle pain (*Zubieta et al., 2001*). The other possibility is that the MORs expressed in GABAergic neurons but not the glutamatergic neurons are constitutively activated during CFA-induced inflammation. The reason that exogenous opioid application did not cause analgesic effect in mice with MORs selectively expressed in GABAergic neurons could be that exogenous opioid activates more MORs than the endogenously active MOR populations, and some of MOR$^+$ neurons targeted by exogenous opioid play opposite roles in pain modulation. This is highly likely because our results showed that the activation of MORs in spinal GABAergic neurons evoked hyperalgesia. In addition, it is worth noting that delta and kappa opioid receptors also play important roles in opioid analgesia (*Corder et al., 2018*). Nevertheless, the distinct mechanism underlying endogenous opioid analgesia provides new opportunity for targeting the pain-gating circuits with new approaches.

## Technical issues

We have developed a new genetic model for studying the functional role of MORs in different populations. This new genetic approach is achieved by inserting a stop cassette flanked by *loxP* between exon 1 and exon 2 of *Oprm1* gene (*Figure 2a*), leading to loss of MOR. This design allows for selectively re-expressing MOR in distinct neuronal population in a Cre-dependent manner. We have verified this strategy at both expressional and functional levels. The key advantage of this strategy is able to maintain the original expression pattern and level of MOR during re-expression, since the re-expression of MORs is still driven by its own promoter. This strategy is different from other studies that employed promoter of other genes in order to selectively express MOR in certain brain areas, which might not restore MOR expression with endogenous pattern (*Cui et al., 2014*).

It is noteworthy that *Oprm1*$^{KI/KI}$ line is not a MOR null mouse line due to the complexity of the *Oprm1* gene. It is known that the *Oprm1* gene has two promoters and produces 3 classes of splicing variants, 7-transmembrane (TM), truncated 6-TM and 1-TM variants (*Pasternak, 2014*). There is also evidence showing that some of them might form heterodimers (*Xu et al., 2013*). Moreover, the regional distribution patterns of these different splicing variants are distinct from one another (*Abbadie et al., 2000a*; *Abbadie et al., 2000b*; *Abbadie et al., 2004*; *Xu et al., 2014*). In generating the *Oprm1*$^{KI/KI}$ mouse line, a stop cassette was inserted after exon 1. As some exon 11 promoter-driving MOR splice variants could skip exon 1 (*Pan et al., 2001*), these MOR splicing variants may still express in *Oprm1*$^{KI/KI}$ mice. However, it has been previously shown that the exon 11-related MOR splicing variants are not involved in morphine analgesia (*Pan et al., 2009*; *Schuller et al., 1999*), consistent with our results showing that morphine analgesia was lost in *Oprm1*$^{KI/KI}$ mice. Therefore, the residual exon 11-related MORs in *Oprm1*$^{KI/KI}$ mice would not affect our conclusion.

In summary, our study deciphered the mechanism underlying opioid analgesia with pharmacological and genetic approaches. We revealed that analgesia by exogenous and endogenous opioids are mediated by MORs expressed in different population of neurons. Our results also demonstrate that MORs in glutamatergic and GABAergic neurons at different areas in the nervous system play distinct roles in modulating nociceptive information processing. This study provides new mechanistic insight into the neural mechanisms underlying pain modulation, paving the way for designing new strategy for pain management.

## Materials and methods

### Key resources table

| Reagent type (species) or resource | Designation | Source or reference | Identifiers | Additional information |
|---|---|---|---|---|
| Genetic reagent (*Mus. musculus*) | *Oprm1*$^{KI/KI}$ | This paper | | See *Figure 2a* |

*Continued on next page*

*Continued*

| Reagent type (species) or resource | Designation | Source or reference | Identifiers | Additional information |
|---|---|---|---|---|
| Genetic reagent (*Mus. musculus*) | *Oprm1*<sup>fl/fl</sup> | This paper and *Zhang et al., 2020* | | See *Figure 3—figure supplement 2a* |
| Genetic reagent (*Mus. musculus*) | MOR-iCreER$^{T2}$ | This paper | | See *Figure 7—figure supplement 1a* |
| Genetic reagent (*Mus. musculus*) | B6.Cg-Tg(Nes-cre)1Kln/J (Nestin-Cre) | Jackson Laboratory | Stock#: 003771 RRID:MGI:2174506 | |
| Genetic reagent (*Mus. musculus*) | STOCK *Slc17a6*$^{tm2(cre)Lowl}$/J (Vglut2-ires-Cre) | Jackson Laboratory | Stock#: 016963 RRID:MGI:5300532 | |
| Genetic reagent (*Mus. musculus*) | STOCK *Slc32a1*$^{tm2(cre)Lowl}$/J (Vgat-ires-Cre) | Jackson Laboratory | Stock#: 016962 RRID:MGI:5300525 | |
| Genetic reagent (*Mus. musculus*) | B6.Cg-*Gt(ROSA)26Sor*$^{tm9(CAG-tdTomato)Hze}$/J (Ai9) | Jackson Laboratory | Stock#: 007909 RRID:MGI:3813511 | |
| Genetic reagent (*Mus. musculus*) | Lbx1-Cre | *Sieber et al., 2007* | | Dr. Yang Liu (Hangzhou Normal University) |
| Genetic reagent (*Mus. musculus*) | SNS-Cre | *Agarwal et al., 2004* | | Dr. Rohini Kuner (University of Heidelberg) |
| Genetic reagent (*Mus. musculus*) | C57BL/6J | SLAC Laboratory | | |
| Ggenetic reagent (*Dependoparvovirus*) | AAV2/9-hSyn-DIO-GCaMP6s-EYFP | Shanghai Taitool Inc | | $4.60 \times 10^{12}$ v.g./ml |
| Genetic reagent (*Dependoparvovirus*) | AAV2/9-Ef1α-DIO-EYFP | Shanghai Taitool Inc | | $3.83 \times 10^{12}$ v.g./ml |
| Genetic reagent (*Dependoparvovirus*) | AAV2/8-hSyn-Cre-EGFP | Shanghai Taitool Inc | | $5.99 \times 10^{12}$ v.g./ml |
| Genetic reagent (*Dependoparvovirus*) | AAV2/8-hSyn-EGFP | Shanghai Taitool Inc | | $4.66 \times 10^{12}$ v.g./ml |
| Antibody | Anti-Mu Opioid Receptor antibody (Rabbit Monoclonal) | Abcam | Cat#: ab134054 | 1:500 for brain and spinal cord, 1:1000 for DRG |
| Antibody | Anti-CGRP (Goat Polyclonal) | Abcam | Cat#: ab36001 RRID:AB_725807 | 1:500 |
| Antibody | anti-c-Fos (Rabbit Polyclonal) | Synaptic System | Cat#: 226003 RRID:AB_2231974 | 1:15000 |
| Antibody | FITC-IB4 | Sigma | Cat#: L2895 | 1:200 |
| Antibody | Anti-GAD67 (Mouse Monoclonal) | Millipore | Cat#: MAB5406 RRID:AB_2278725 | 1:2000 |
| Antibody | Anti-VGluT2 (Guinea pig Polyclonal) | Millipore | Cat#: AB2251-I RRID:AB_2665454 | 1:500 |
| Antibody | Cy5 AffiniPure Donkey Anti-Goat IgG (H+L) | Jackson Immuno Research Laboratories | Cat#: 705-175-147 RRID:AB_2340415 | 1:200 |
| Antibody | Cy3 AffiniPure Donkey Anti-Rabbit IgG (H+L) | Jackson Immuno Research Laboratories | Cat#: 711-165-152 RRID:AB_2307443 | 1:200 |
| Antibody | Alexa Fluor 488 AffiniPure Donkey Anti-Rabbit IgG (H+L) | Jackson Immuno Research Laboratories | Cat#: 711-545-152 RRID:AB_2313584 | 1:200 |
| Antibody | Biotin-SP AffiniPure Goat Anti-Rabbit Polyclonal IgG (H+L) | Jackson Immuno Research Laboratories | Cat#: 111-065-003 RRID:AB_2337959 | 1:200 |

*Continued on next page*

*Continued*

| Reagent type (species) or resource | Designation | Source or reference | Identifiers | Additional information |
|---|---|---|---|---|
| Antibody | Alexa Fluor 488 AffiniPure Donkey Anti-Mouse IgG (H+L) | Jackson Immuno Research Laboratories | Cat#: 715-545-150 RRID:AB_2340846 | 1:200 |
| Antibody | Alexa Fluor 647 AffiniPure Donkey Anti-Guinea Pig IgG (H+L) | Jackson Immuno Research Laboratories | Cat#: 706-605-148 RRID:AB_2340476 | 1:200 |
| Sequence-based reagent | RNAscope Probe-*Oprm1*-O3 | Advanced Cell Diagnostics | Cat#: 493251 | |
| Sequence-based reagent | RNAscope Probe-*tdTomato*-C3 | Advanced Cell Diagnostics | Cat#: 317041 | |
| Sequence-based reagent | RNAscope Probe-*Slc32a1*-C2 (Vgat) | Advanced Cell Diagnostics | Cat#: 319191 | |
| Sequence-based reagent | RNAscope Probe-*Slc17a6*-C3 (Vglut2) | Advanced Cell Diagnostics | Cat#: 319171 | |
| Sequence-based reagent | RNAscope Probe-*Slc17a7*-C3 (Vglut1) | Advanced Cell Diagnostics | Cat#: 416631 | |
| Sequence-based reagent | RNAscope Probe- 3-Plex Negative Control | Advanced Cell Diagnostics | Cat#: 320871 | |
| Commercial assay or kit | VECTASTAIN ABC-Peroxidase Kit | Vector | Cat#: PK-4000 RRID:AB_2336818 | 1:100 |
| Commercial assay or kit | RNAscope Multiplex Fluorescent Reagent Kit v2 | Advanced Cell Diagnostics | Cat#: 323100 | |
| Chemical compound, drug | CFA | Sigma | Cat#: F5881 | 50% (vol/vol) |
| Chemical compound, drug | Formaldehyde | Sigma | Cat#: F1635 | 5% (vol/vol) |
| Chemical compound, drug | Tamoxifen | Sigma | Cat#: T5648 | 150 mg/kg |
| Chemical compound, drug | β-FNA | Tocris | Cat#: 0926 | 10 mg/kg |
| Chemical compound, drug | Red beads | Lumaflour | Cat#: Retrobeads IX | 1:10 diluted in PBS |
| Software, algorithm | Image J | NIH | | |
| Software, algorithm | LabState | Anilab | | |
| Software, algorithm | Fscope | BiolinkOptics | | |

## Animals

Male *Oprm1*$^{KI/KI}$, *Oprm1*$^{fl/fl}$, MOR-iCreER$^{T2}$, Nestin-Cre (JAX003771), Vglut2-ires-Cre (JAX016963, referred to as Vglut2-Cre), Vgat-ires-Cre (JAX016962, referred to as Vgat-Cre), Ai9 (*Rosa26*$^{tdTomato}$, JAX007909), Lbx1-Cre (**Sieber et al., 2007**), SNS-Cre (**Agarwal et al., 2004**) and C57BL/6J wild-type mice were used for experiments. Three to six mice were raised in each cage. We generated MOR conditional re-expression mouse line, referred to as *Oprm1*$^{KI/KI}$. *Oprm1* allele of *Oprm1*$^{KI/KI}$ mice harbors two *loxP* sites flanking one stop cassette between *Oprm1* exon 1 and exon 2 (**Figure 2a**), which allows for conditional manipulation of MOR in neurons with Cre recombinase by using Cre/*loxP* strategy. *Oprm1*$^{KI/KI}$ mice were crossed with Nestin-Cre, Vglut2-Cre, Vgat-Cre, Lbx1-Cre and SNS-Cre mice to get the MOR conditional knock-in mice, referred to as Nestin-Cre/*Oprm1*$^{KI/KI}$, Vglut2-Cre/*Oprm1*$^{KI/KI}$, Vgat-Cre/*Oprm1*$^{KI/KI}$, Lbx1-Cre/*Oprm1*$^{KI/KI}$ and SNS-Cre/*Oprm1*$^{KI/KI}$. For *Oprm1*$^{fl/fl}$ mouse line (**Zhang et al., 2020**), *Oprm1* allele of *Oprm1*$^{fl/fl}$ mice harbors two *loxP* sites flanking *Oprm1* exon 2 and exon 3 (**Figure 3—figure supplement 2a**), which allows for conditional deletion of MOR in neurons with Cre recombinase. *Oprm1*$^{fl/fl}$ mice were crossed with Vgat-Cre, Vglut2-Cre and Lbx1-Cre mice to get the MOR conditional knockout mice. For MOR-iCreER$^{T2}$ mouse line, a CreER element was inserted into exon 2 (**Figure 7—figure supplement 1a**).

All mice were raised on a 12 hr light/dark cycle (lights on at 7:00 a.m.) with ad libitum food and water. Behavioral experiments were carried out in the light phase. All procedures were approved by the Animal Care and Use Committee of the Institute of Neuroscience, Chinese Academy of Sciences, Shanghai, China.

### Tissue preparation

Mice were anesthetized with pentobarbital sodium (100 mg/kg) and perfused transcardially with saline followed by 4% paraformaldehyde (PFA, Sigma). The brain, DRG (lumber segments), and spinal cord (lumbar segments) were dissected and post-fixed overnight at 4°C in 4% PFA, followed by dehydration in 30% sucrose dissolved in PBS at 4°C. Sections (40 µm for brain, 30 µm for spinal cord, 10 µm for DRG, and 20 µm for in situ hybridization) were prepared with a cryostat (Leica CM 1950) for immunostaining or in situ hybridization.

### Immunofluorescent staining

Immunofluorescent staining was performed as described previously (Gao et al., 2019). Tissue sections were blocked for 1 hr at room temperature in PBST (0.3% Triton X-100) with 5% normal donkey serum, followed by incubation with primary antibodies at 4°C overnight and secondary antibodies at room temperature for 2 hr in PBST (0.3% Triton X-100) with 1% normal donkey serum. Primary antibodies used for immunofluorescent staining were anti-MOR (rabbit, 1:500 for brain and spinal cord, 1:1000 for DRG, ABCAM, ab134054), anti-CGRP (goat, 1:500, ABCAM, ab36001), FITC-IB4 (1:200, Sigma, L2895), anti-c-Fos (rabbit, 1:15000, Synaptic System, 226003), anti-VGluT2 (guinea pig, 1:2000, Millipore, AB2251-I), anti-GAD67 (mouse, 1:500, Millipore, MAB5406). Secondary antibodies were donkey anti-goat IgG-Cy5, donkey anti-rabbit IgG-Cy3, donkey anti-rabbit IgG-Alexa 488, donkey anti-mouse IgG-Alexa 488 and donkey anti-guinea pig IgG-Cy5 (1:200, Jackson ImmunoResearch Laboratories).

### DAB staining

Tissue sections were blocked in PBS with 0.3% $H_2O_2$ for 30 min, and then blocked in PBST (0.3% Triton X-100) with 1% goat serum for 2 hr at room temperature. Next, sections were incubated with primary antibody of anti-MOR (rabbit, 1:500, ABCAM, ab134054) for 36 hr at 4°C, followed by secondary antibody of Biotin-SP goat anti-rabbit IgG (1:200, Jackson ImmunoResearch Laboratories) for 2 hr at room temperature. After incubation with ABC (1:100, Vector, PK-4000) for 1 hr, sections were developed in DAB (3, 3-Diaminobenzidine)-ammonium nickel sulfate developing solution. In the end, ethyl alcohol and xylene were used for decoloration and sections were mounted with quick-hardening mounting medium. For quantitative analysis of the expression level of MOR, sections from the test group were stained with those of wild-type control in one well to control the variables.

### In situ hybridization

To investigate the identity of MOR-expressing neurons, multiple in situ hybridization experiments were performed using RNAscope Fluorescent Multiplex Assay with Oprm1, Slc17a6, Slc32a1, Slc17a7 probes (Advanced Cell Diagnostics), and the 3-Plex negative probe was used as control. Tissue sections were spread on slides and heated for 2 hr at 60°C, and then kept at −80°C before experiments. According to manufacturer's protocol, slides were pretreated with hydrogen peroxide for 20 min at room temperature and washed in DEPC-$H_2O$ for 1 min. Then slides were transferred to boiling retrieval regent for 7 min, and rinsed once in DEPC-$H_2O$ at room temperature. Protease digestion was performed for 15 min in 40°C HybEZ oven. After washed in DEPC-PBS for 3 min and rinsed in DEPC-$H_2O$, slides were treated with ethanol for 3 min twice, followed by air-dry at room temperature. Pre-warmed probes were mixed and slides were hybridized for 2 hr in 40°C HybEZ oven. Signal amplification fluorescent labels are TSA-based. The specificity of MOR-iCreER$^{T2}$ was validated using RNAscope assay in MOR-iCreER$^{T2}$ × Ai9 mice with Oprm1 and tdTomato probes with the same procedure.

### Image acquisition and analysis

Images were taken using Olympus VS120 microscope, Olympus FV3000 confocal fluorescence microscope and Nikon Tie-A1 plus confocal fluorescence microscope. For co-localization analysis, confocal

images taken by Olympus FV3000 were cropped in 400 μm × 400 μm areas. The number of counting areas ranges from 1 to 6, which depends on the area of each brain region in one section. Cell counting was carried out manually using Fiji (NIH). For the calculating of MOR re-expression efficiency using MOR DAB staining, images were taken by Olympus VS120 microscope and processed in Fiji. Images were transformed to 8-bit and substrated the background, and then gray value in specific brain regions were measured in 800 μm × 800 μm areas, except for the CeA that 600 μm × 600 μm areas were used in Vgat-Cre/*Oprm1*$^{KI/KI}$ group. The expression level of MOR was normalized to the mean gray value of MOR signal in each brain region of wild-type mice.

## Stereotaxic injection

To do the stereotaxic injection, mice were anesthetized with pentobarbital sodium (100 mg/kg) and then hold on stereotaxic apparatus. The skull was exposed by midline scalp incision, and craniotomy was performed for introduction of a microinjection glass pipette into brain. The craniotomy windows (~1.5 mm diameter) were made using a hand-held drill over the target areas. To record the neuronal activity of MOR$^+$ neurons during noxious stimuli, AAV-hSyn-DIO-GCaMP6s-EYFP (AAV 2/9, titer: 4.60 × 10$^{12}$ v.g./ml, 300 nl, Shanghai Taitool Inc) or AAV-Ef1α-DIO-EYFP (AAV 2/9, titer: 3.83 × 10$^{12}$ v.g./ml, 300 nl, Shanghai Taitool Inc) virus was injected to right PBN of MOR-iCreER$^{T2}$ mice. Optical fiber terminal was implanted 0.1–0.15 mm upper to the injection site. Tamoxifen was applied 1 week after surgery. To re-express the MOR in PBN, AAV-hSyn-Cre-EGFP (AAV 2/8, titer: 5.99 × 10$^{12}$ v.g./ml, 300 nl, Shanghai Taitool Inc) or AAV-hSyn-EGFP (AAV 2/8, titer: 4.66 × 10$^{12}$ v.g./ml, 300 nl, Shanghai Taitool Inc) virus was bilaterally injected to the PBN of *Oprm1*$^{KI/KI}$ mice. The coordinates for PBN are 5.10 mm posterior to bregma, ± 1.65 mm lateral to midline, 3.65 mm ventral to skull surface.

## Pain behavior tests

To evaluate morphine-induced analgesia, morphine was diluted in saline and injected subcutaneously (10 mg/kg) or intrathecally (1.0 nmol, 5 μl), and behavioral tests were performed 40 min or 30 min later, respectively. For tail immersion test, mice were gently restrained in a cotton towel and one third of the tail from tip was then dipped into a water bath of 48 or 50℃. And the tail flick latency was recorded, with a cut-off time of 20 s to avoid tissue damage.

The hot plate test was performed by placing mice on the hot plate (Ugo Basile, Comerio, Italy) at 52℃, and the first appearance of hind paw licking, hind paw lifting or jumping was recorded. A maximal cut-off of 45 s was set to avoid tissue damage.

For testing mechanical sensitivity, mouse hind paw was perpendicularly stimulated with a series of von Frey hairs with logarithmically incrementing stiffness (0.07–1.4 grams, Stoelting, Wood Dale, IL). One filament was applied five times in one test, with intervals more than 5 s, and the paw withdrawal times were recorded. Paw withdrawal frequency was calculated to present the mechanical threshold.

## Formalin test

For the formalin test, the mice received intraplantar injection of formalin (Sigma, F1635, 5%, vol/vol diluted in saline, 10 μl) and then replaced in transparent plastic chambers. Morphine (10 mg/kg) or saline was injected subcutaneously or intraperitoneally 30 min before formalin injection. Total duration of the animal spending in licking and flinching behaviors of the injected paw was counted manually in 5 min interval for 1 hr. The first phase of nociceptive responses was calculated during 0–10 min, and the second phase of nociceptive responses was calculated during 10–60 min.

## Complete Freund's adjuvant (CFA)-induced inflammatory pain

For the CFA-induced pain, the mice were subcutaneously injected of 50% CFA (20 μl) into the plantar of the right hind paws. The CFA (50%) was prepared by isometric CFA (Sigma, F5881) and saline, and the mixture was emulsified by intensive mixing. The mechanical threshold and thermal sensitivity on the ipsilateral paw during chronic pain were tested by von Frey test and plantar test, respectively. For the von Frey test, the modified Dixon's up-down method (*Chaplan et al., 1994*) was conducted by using a series of von Frey filaments to determine the 50% hind paw withdrawal threshold. For the plantar test, the Hargreaves apparatus (Ugo Basile, Comerio, Italy) was used to measure the paw

withdrawal latency, a 20 s was set as the cut-off time to prevent injury. Both the thermal threshold and mechanical sensitivity were averaged from two trials, separated by 30 min intervals at each time point. To determine the morphine analgesia on CFA-induced inflammatory pain, morphine (10 mg/kg) was subcutaneously administrated on day 7 and day 14 after CFA injection, and then the mechanical threshold and thermal sensitivity were measured on 30, 60, 90, 120, 180 and 240 min, respectively. Both the mechanical threshold and thermal sensitivity were measured once at each time point for morphine test. To investigate the endogenous opioid function mediated by MOR expressed on the GABAergic neurons, β-funaltrexamine hydrochloride (β-FNA, 10 mg/kg), a specific MOR agonist, was intraperitoneally injected on day 42 after the tests on pain sensitivity in Vgat-Cre/$Oprm1^{KI/KI}$ and $Oprm1^{KI/KI}$ mice. The mechanical threshold was tested on the ipsilateral paw before and 15, 30, 45, 60, 90, 120, 150, 180 and 210 min after the β-FNA injection.

## Open field test

Locomotor activity of mice was evaluated by open field test. Mice were placed in the testing room for about 1 hr for habituation. For baseline test, mice were placed in the open field boxes (40 × 40 × 40 cm) and videotaped individually. Then mice were taken out from the boxes and subcutaneously injected with 10 mg/kg morphine. The open field test was performed 40 min after injection. Total travel distance and average speed were recorded in a 10 min period. The track was analyzed by LabState (AniLab).

## Tamoxifen induction

To induce the expression of Cre recombinase in MOR-iCreER$^{T2}$ mice with AAV-hSyn-DIO-GCaMP6s-EYFP or AAV-Ef1α-DIO-EYFP virus, tamoxifen (20 mg/ml, Sigma, T5648) was dissolved in sunflower oil at room temperature (Guenthner et al., 2013), and intraperitoneally injected to MOR-iCreER$^{T2}$ mice 7 days after virus injection for 4–5 consecutive days (150 mg/kg).

## Fiber photometry recording

For recording the neuronal response to acutely nociceptive tail pinch stimulation in freely-moving animals, mice were placed on an open field and habituated for 3 min while the implanted optic fiber was connected to an external patch cord. Unexpected tail pinch using a clip (~10 s) was delivered for 3–5 times with an interval of 60–120 s. The start and stop time of each stimulation was tagged by triggering TTL signal (Isolated Pulse Stimulator Model 2100, A-M SYSTEMS), which was synchronously output to the fiber photometry system, and recorded with calcium signal simultaneously at 1000 Hz using F-scope-G-2 (Biolink Optics). To examine the morphine's effect on PBN MOR$^+$ neurons during tail pinch, tail pinch was first applied for three times to record a basal response. Morphine (10 mg/kg) was injected intraperitoneally, then tail pinch and recording were performed 40 min later. Based on our data, morphine lost its analgesic effect in about 4 hr, thus we measured the tail pinch-induced response 5 hr post morphine application. For recording the neuronal response during nociceptive hindpaw pinch and hot water, anesthetized animals (1% pentobarbital sodium, 50 mg/kg) received right hindpaw pinch or tail immersion (40 or 52℃) for three times for each stimulation (~10 s) with an interval of ~120 s. The excitation laser power was controlled at ~15 μW. The system noise was recorded after disconnecting the tip of photometry fiber from implanted fiber and blocking any optic input. Data were transformed into mat file and analyzed in MATLAB.

## Fiber photometry data analysis

The onset of event stimulation was identified first. Fluorescence values were low-pass filtered at 2 Hz using a 4th order Butterworth filter with zero-phase distortion and then subtracted the noise signal of recording system. The resulting values were aligned to event onset and corresponding fluorescence values in each stimulation was derived. The dynamics of fluorescence in each stimulation was calculated by

$$\Delta F/F = (F - F_0)/F_0$$

F stands for each fluorescence value and $F_0$ stands for the median of fluorescence values during baseline window (−5 to 0 s before stimulation). The averaged fluorescence change was visualized by heat plot using jet colormap in MATLAB. To quantify the change of fluorescence values across

stimulation period, the event windows were defined (0–5 s after the onset of tail pinch). The averaged fluorescence changes in each baseline and event window was calculated and compared.

### c-Fos double-staining with retro-beads

One week after red retro-beads injection in PBN, c-Fos experiment was performed in Vgat-Cre/*Oprm1*KI/KI mice. Mice were placed in the testing room for about 2 hr for habituation. After habituation, morphine (1.0 nmol/ 5 μl) or saline was intrathecally injected to Vgat-Cre/*Oprm1*KI/KI mice, and these mice were sacrificed 2 hr after morphine injection for further staining. Then immunostaining of c-Fos (rabbit, 1:15000, Synaptic System, 226003) was performed with the method of immunofluorescent staining.

### Morphine-induced nocifensive behaviors

The mice were placed in transparent plastic chambers on a glass board. Morphine (1 nmol/ 5 μl) was injected intrathecally and the performance of mice after morphine administration was recorded by camera. Time spent on grooming, hindpaw licking, flinching, and body twisting in the following 30 min was analyzed in Matlab R2009a manually.

### Quantification and statistical analysis

Statistical analysis was performed using GraphPad Prism 6, MATLAB R2009a and MATLAB R2015b. All the experiments were performed in the blinded manner. For behavioral tests, experimenters were blinded to the mouse genotypes or virus information. The number of experiments performed with independent mice (n) is indicated in the legends. The data were analyzed using Repeated-measures two-way ANOVA, one-way ANOVA followed by Bonferroni post hoc analysis, and two-tailed un-paired student's *t*-test. The cut-off for significance was held at p=0.05. All data are presented as mean with SEM.

## Acknowledgements

We thank Dr. Mu-Ming Poo for comments on the manuscript. We thank all the lab members of Y-GS for helpful discussion. We thank Dr. Rohini Kuner (University of Heidelberg) for providing the SNS-Cre mouse line. This work was supported by the National Natural Science Foundation of China (Grant No. 31825013, 31800877, 61890952), Shanghai Municipal Science and Technology Major Project (Grant No. 2018SHZDZX05), and the Strategic Priority Research Program of the Chinese Academy of Sciences (Grant No. XDB32010200). Y-ND is supported by China Postdoctoral Science Foundation (Grant No. 2018M640426) and the Shanghai Post-doctoral Excellence Program (Grant No. 2018038).

## Additional information

### Funding

| Funder | Grant reference number | Author |
| --- | --- | --- |
| Shanghai Municipal Science and Technology Major Project | 2018SHZDZX05 | Yan-Gang Sun |
| Chinese Academy of Sciences | XDB32010200 | Yan-Gang Sun |
| China Postdoctoral Science Foundation | 2018M640426 | Yan-Nong Dou |
| Shanghai Postdoctoral Excellence Program | 2018038 | Yan-Nong Dou |
| National Natural Science Foundation of China | 31825013 | Yan-Gang Sun |
| National Natural Science Foundation of China | 31800877 | Yan-Nong Dou |
| National Natural Science Foundation of China | 61890952 | Yan-Gang Sun |

The funders had no role in study design, data collection and interpretation, or the decision to submit the work for publication.

### Author contributions
Xin-Yan Zhang, Formal analysis, Investigation, Writing - original draft, Writing - review and editing; Yan-Nong Dou, Formal analysis, Funding acquisition, Investigation, Writing - original draft, Writing - review and editing; Lei Yuan, Formal analysis, Methodology; Qing Li, Yan-Jing Zhu, Investigation; Meng Wang, Investigation, Methodology; Yan-Gang Sun, Conceptualization, Funding acquisition, Project administration, Writing - review and editing

### Author ORCIDs
Xin-Yan Zhang (iD) https://orcid.org/0000-0001-8171-2007
Yan-Nong Dou (iD) https://orcid.org/0000-0002-9096-6284
Yan-Gang Sun (iD) https://orcid.org/0000-0003-0768-7733

### Ethics
Animal experimentation: This study was performed in strict accordance with the recommendations in the Guide for the Care and Use of Laboratory Animals of the Institute of Neuroscience, CAS. All procedures were approved by the Animal Care and Use Committee of the Institute of Neuroscience, Chinese Academy of Sciences, Shanghai, China (Protocol number: NA-005-2019).

### Decision letter and Author response
Decision letter https://doi.org/10.7554/eLife.55289.sa1
Author response https://doi.org/10.7554/eLife.55289.sa2

## Additional files

### Supplementary files
- Source code 1. Code for scorevideo.
- Source code 2. Code for fiber photometry.
- Transparent reporting form

### Data availability
All data generated or analysed during this study are included in the manuscript and supporting files.

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
