## [Decision Letter]

**Acceptance summary:**

This manuscript addresses important questions in the field of opioids, pain, and analgesia by delineating the relative contributions of µ-opioid receptors in specific neurochemically-distinct cell types. The study includes an impressive array of methods and cutting-edge tools, including mouse genetics, in vivo activity imaging, pharmacology, and behavior. The results are important for understanding how µ-opioid receptors contribute to analgesia induced by exogeneous (e.g., morphine) and endogenous opioids.

**Decision letter after peer review:**

Thank you for submitting your article "Different neuronal populations mediate inflammatory pain analgesia by exogenous and endogenous opioids" for consideration by *eLife*. Your article has been reviewed by three peer reviewers, one of whom is a member of our Board of Reviewing Editors, and the evaluation has been overseen by Kate Wassum as the Senior Editor. The following individual involved in review of your submission has agreed to reveal their identity: Jose A Moron (Reviewer #3).

The reviewers have discussed the reviews with one another and the Reviewing Editor has drafted this decision to help you prepare a revised submission.

Summary:

This manuscript addresses important questions in the field of opioids, pain, and analgesia by delineating the relative contributions of µ-opioid receptors (MORs) in specific neurochemically-distinct cell types (glutamatergic and GABAergic cells). The study includes an impressive array of methods and cutting-edge tools, including mouse genetics, in vivo activity imaging, pharmacology, and behavior. The results show that MORs expressed in glutamatergic neurons play a large role in mediating analgesia by exogenous morphine while MORs expressed in GABAergic neurons are important for analgesia by endogenous opioids during chronic inflammatory pain.

Essential revisions:

1) In the reporter line expressing dtTomato via a Cre-lox approach under the control of the *Oprm1* promoter only 46% of MOR-expressing neurons express the reporter td-Tomato. The low efficiency brings in a large element of ambiguity and uncertainty in the inferences made using this method regarding neuronal populations mediating effects of MOR with respect to analgesia. Since the authors then employed *Oprm1* in situ hybridization for characterizing expression of MOR in neuronal populations, the data from the reporter line are redundant and could be misleading. We suggest focusing on the in situ data and eliminating Figure 1A-C.

2) Specificity controls for the in situ hybridization probes and conditions applied are missing or at least not readily visible in the main figures or supplementary figures. The same applies to the multiple antibodies employed for immunohistochemistry. Overall, this entire section on expression analyses is quite superficial and the reviewer is not convinced about its value.

3) The authors then created a knock-in line, which makes MOR re-expression Cre-dependent manner in a knockout-like background (KI mice). Because these mice were employed in functional analyses to ascribe specific opioidergic functions to MOR expression in specific subsets of neurons, it is crucial to show the extent to which MOR re-expression follows the endogenous cellular pattern of MOR expression, ideally this will be close to the endogenous expression level and if not this limitation should be thoroughly discussed. Experiments showing specificity of re-expression of MOR in an Opmr1-specific manner (via co-in situs for Opmr1) and in the corresponding Cre-line-specific manner (co-immunohistochemistry with corresponding markers) are needed. Moreover, quantitative analyses via Western blotting need to be demonstrated in each of these lines.

4) There are excitatory neurons which lack Vglut2 and express Vglut1. Here, Vglut2-Cre mice were used for re-expression of MOR, and used to derive inferences on roles of MOR in excitatory neurons. If the contribution of MORs expressed in the Vglut1 population cannot be experimentally ruled out, then the interpretation needs to be changed in light of this and this important point needs to be addressed.

5) Opioidergic drugs not only inhibit the sensory component of pain, but very importantly also impact on the emotional-motivational component of pain and spontaneous pain. The latter was not adequately addressed here. The reviewer missed behavioral experiments that address the latter in the mouse lines studied. This can be remedied either by examining pain in the context of several well-established behavioral tests to address this, or by altering the interpretation of the data and clearly discussing that these data only relate to the sensory component of pain.

6) One issue that is unresolved is the contribution of δ and kappa opioid receptors that are also potential sites of analgesic action, this limitation should be addressed.

7) Another is the surprising expression of Vgat in sensory neurons that are supposed to be glutamatergic (Du et al., 2017) This should be discussed as it confuses the interpretation of the data.

8) Figure 3I and J, can the authors discuss the rationale behind doing the morphine test 1 week after the post-CFA sensitivity test. For example, in Figure 3I, post-CFA sensitivity test is done on day 1 but morphine test is done on day 7. Similarly, with Figure 3J, post-CFA sensitivity test is done on day 7 and morphine test is done on day 14.

9) Figure 6—figure supplement 1J and K, the authors should provide clarification as to why the time points selected for von Frey and Hargreaves test different from the rest of the experiments in the paper.

[Editors' note: further revisions were suggested prior to acceptance, as described below.]

Thank you for resubmitting your work entitled "Different neuronal populations mediate inflammatory pain analgesia by exogenous and endogenous opioids" for further consideration by *eLife*. Your revised article has been evaluated by Kate Wassum as the Senior and Reviewing Editor and one peer reviewer.

The manuscript has been improved but there are some remaining issues that need to be addressed before acceptance, as outlined below:

We agreed that the manuscript is much improved and is a very nice piece of work. But we have one remaining concern, and a few additional suggestions that need to be addressed.

Essential Revision:

The N for Figures 3C and 4C is low, and the variability is high. Thus, a higher N is needed to support the statement (and related statements for Figure 4: "Immunostaining showed the expression of MOR in the brain areas composed mostly by glutamatergic neurons, such as the habenula, thalamus and PBN, was largely restored in Vglut2-Cre/*Oprm1*^KI/KI^ mice (Figure 3B-C, Figure 3—figure supplement 1A-B)". This would be the ideal approach. If it not possible, then these claims need to be tempered.

---

## [Author Response]

Essential revisions:1) In the reporter line expressing dtTomato via a Cre-lox approach under the control of the Oprm1 promoter only 46% of MOR-expressing neurons express the reporter td-Tomato. The low efficiency brings in a large element of ambiguity and uncertainty in the inferences made using this method regarding neuronal populations mediating effects of MOR with respect to analgesia. Since the authors then employed Oprm1 in situ hybridization for characterizing expression of MOR in neuronal populations, the data from the reporter line are redundant and could be misleading. We suggest focusing on the in situ data and eliminating Figure 1A-C.

As suggested by the reviewers, we have deleted these data from our manuscript.

2) Specificity controls for the in situ hybridization probes and conditions applied are missing or at least not readily visible in the main figures or supplementary figures. The same applies to the multiple antibodies employed for immunohistochemistry. Overall, this entire section on expression analyses is quite superficial and the reviewer is not convinced about its value.

Due to the possible noise of TSA-based signal amplification in RNAscope assay, we have performed control experiment by using the 3-Plex negative control probe. We have applied the same imaging and adjusting parameters between experimental group and control group, and the results showed that no signal was detected in the negative probe group, indicating the specificity of signals we detected. And we have now included these results in the supplementary figures. Please see Figure 1—figure supplement 1A-B. In this part of work, we have provided new information about the identity of MOR^+^ neurons in several brain regions.

The anti-MOR antibody was commonly used in previous studies (Wang et al., 2018; Cui et al., 2014), and we have used the MOR KO mice to further confirm its specificity in our experiments.

3) The authors then created a knock-in line, which makes MOR re-expression Cre-dependent manner in a knockout-like background (KI mice). Because these mice were employed in functional analyses to ascribe specific opioidergic functions to MOR expression in specific subsets of neurons, it is crucial to show the extent to which MOR re-expression follows the endogenous cellular pattern of MOR expression, ideally this will be close to the endogenous expression level and if not this limitation should be thoroughly discussed. Experiments showing specificity of re-expression of MOR in an Opmr1-specific manner (via co-in situs for Opmr1) and in the corresponding Cre-line-specific manner (co-immunohistochemistry with corresponding markers) are needed. Moreover, quantitative analyses via Western blotting need to be demonstrated in each of these lines.

As suggested by the reviewers, we have examined the MOR expression with the neuronal markers, such as the Vglut2 and GAD67. Co-immunostaining of MOR with Vglut2 or GAD67 showed that the re-expressed MORs in Vglut2-Cre/*Oprm1*^KI/KI^ and Vgat-Cre/*Oprm1*^KI/KI^ mice distributed in the endogenous pattern. We have included these data in the Supplementary figures. Please see Figure 3—figure supplement 1A-B and Figure 4—figure supplement 1A-B.

We have tried the western blotting experiment with our MOR-rescued mice. However, the MOR antibody did not work well in this assay, as shown in literature (Lupp et al., Regul Pept 2011). We thus did not further pursue this experiment, but analyzed the intensity of the MOR immunostaining signals instead.

4) There are excitatory neurons which lack Vglut2 and express Vglut1. Here, Vglut2-Cre mice were used for re-expression of MOR, and used to derive inferences on roles of MOR in excitatory neurons. If the contribution of MORs expressed in the Vglut1 population cannot be experimentally ruled out, then the interpretation needs to be changed in light of this and this important point needs to be addressed.

We agree with the reviewers that excitatory neurons are composed of several subpopulations. MORs are likely expressed in non-Vglut2 excitatory neurons, and these MORs might be also involved in opioid analgesia. We have revised our statement to specify the role of the MORs expressed in VGluT2^+^ glutamatergic neurons in morphine analgesia. In the Discussion, we also discussed the possible role of MORs expressed in other subtypes of glutamatergic neurons. Please see the first paragraph of the subsection “Neural mechanisms underlying morphine analgesia”.

5) Opioidergic drugs not only inhibit the sensory component of pain, but very importantly also impact on the emotional-motivational component of pain and spontaneous pain. The latter was not adequately addressed here. The reviewer missed behavioral experiments that address the latter in the mouse lines studied. This can be remedied either by examining pain in the context of several well-established behavioral tests to address this, or by altering the interpretation of the data and clearly discussing that these data only relate to the sensory component of pain.

We agree with the reviewers. Previous studies have suggested the essential role of opioid drugs for the affective component of pain. In our manuscript, we have only performed pain tests that reveal the antinociceptive effect of morphine in sensory component of pain. We have revised the manuscript to indicate that our study only addressed the role of MOR in the sensory component of pain. Please see the subsection “Neural mechanisms underlying morphine analgesia”.

6) One issue that is unresolved is the contribution of δ and kappa opioid receptors that are also potential sites of analgesic action, this limitation should be addressed.

We agree with the reviewer that δ and kappa opioid receptors are also potential sites for opioid drugs. In our manuscript, only mu opioid receptors are studied, thus we have discussed this in the revised text. Please see the subsection “Analgesia by endogenous opioids in chronic inflammatory pain”.

7) Another is the surprising expression of Vgat in sensory neurons that are supposed to be glutamatergic (Du et al., 2017) This should be discussed as it confuses the interpretation of the data.

Due to the surprising expression of Vgat in primary neurons, it is possible that MORs expressed in the Vgat^+^ neurons of DRG. If it is true, these MORs might involve in sensory transmission. However, we observed no significant MOR signal in the DRG of Vgat-Cre/*Oprm1*^KI/KI^ mice, which have restored MORs in Vgat^+^ neurons. Although we can’t exclude the limitation of sensitivity for immunostaining, this result showed that the MOR barely expressed in Vgat^+^ neurons of DRG. This has been discussed in the second paragraph of the subsection “Neural mechanisms underlying morphine analgesia”.

**Author response image 1. sa2fig1:** MOR and CGRP staining in DRG of several MOR-rescue mouse lines. Scale bar, 100 μm.

8) Figure 3I and J, can the authors discuss the rationale behind doing the morphine test 1 week after the post-CFA sensitivity test. For example, in Figure 3I, post-CFA sensitivity test is done on day 1 but morphine test is done on day 7. Similarly, with Figure 3J, post-CFA sensitivity test is done on day 7 and morphine test is done on day 14.

For CFA-induced chronic inflammatory pain, both mechanical and thermal sensitivities were tested at several time points from 2 h to 7 or 8 weeks after CFA application. To determine morphine analgesia during CFA-induced inflammatory pain, we chose the time points when the mice still exhibited pain hypersensitivity (1-3 weeks after CFA application). Since both mechanical and thermal sensitivities were tested on same cohort of mice, to avoid the possible residual effects of morphine, we tested the mechanical sensitivity using von-Frey test on day 7 after CFA application, and the thermal sensitivity one week later. We have tested the mechanical sensitivity in day 1 for Figure 3J, thus have now included the data for day 1 in Figure 3J.

9) Figure 6—figure supplement 1J and K, the authors should provide clarification as to why the time points selected for von Frey and Hargreaves test different from the rest of the experiments in the paper.

The experiment in the time point of day 7 was conflict with other experimental arrangements, thus the test was performed with one week delay. Although the time points chose in this mouse line were different from the rest of the experiments in the paper, which might be a pity that it is not ideal for the overall experiment design, the present data provide objective evidence showing the morphine analgesia in CFA-induced inflammatory pain.

[Editors' note: further revisions were suggested prior to acceptance, as described below.]

The manuscript has been improved but there are some remaining issues that need to be addressed before acceptance, as outlined below:We agreed that the manuscript is much improved and is a very nice piece of work. But we have one remaining concern, and a few additional suggestions that need to be addressed.Essential Revision:The N for Figures 3C and 4C is low, and the variability is high. Thus, a higher N is needed to support the statement (and related statements for Figure 4: "Immunostaining showed the expression of MOR in the brain areas composed mostly by glutamatergic neurons, such as the habenula, thalamus and PBN, was largely restored in Vglut2-Cre/Oprm1^KI/KI^ mice (Figure 3B-C, Figure 3—figure supplement 1A-B)". This would be the ideal approach. If it not possible, then these claims need to be tempered.

As suggested by the reviewers, we have modified our description for the immunostaining data of MOR re-expression. Please see subsections “MORs in Vglut2^+^ glutamatergic neurons mediate exogenous opioid analgesia” and “MORs in GABAergic neurons mediate endogenous opioid analgesia”.